# Correctness-Aware Knowledge Distillation for Enhanced Student Learning

**Ishan Mishra**                                                                                  *mishra.10@iitj.ac.in*
*Department of Computer Science*
*Indian Institute of Technology Jodhpur*

**Deepak Mishra**                                                                                   *dmishra@iitj.ac.in*
*Department of Computer Science*
*Indian Institute of Technology Jodhpur*

**Jinjun Xiong**                                                                                  *jinjun@buffalo.edu*
*Department of Computer Science and Engineering*
*University at Buffalo*

**Reviewed on OpenReview:** *https://openreview.net/forum?id=XpRXmzd2sF*

## Abstract

In real-world learning, students rely on their mentors for guidance but must also develop the ability to recognize and learn from their mentors' mistakes. Inspired by this mentor-critic dynamic, we propose Mentor-Critic Distillation (MCD), a novel framework for knowledge distillation in machine learning. Traditional distillation methods risk transferring both correct insights and errors from the mentor (teacher model) to the student model, which can hinder student performance. Notably, previous state-of-the-art approaches fail to account for scenarios where the teacher is incorrect, often leaving the student model vulnerable to inheriting these errors. To address this limitation, MCD introduces a weighted knowledge transfer mechanism that decouples the learning process based on the mentor's correctness. When the mentor model is correct, the student model follows the mentor's guidance with a large weight on knowledge transfer. However, when the mentor is incorrect, the student relies more on the ground truth but still learns inter-class relationships from the mentor, adjusting the weight toward task-specific losses such as cross-entropy. This mentor-critic approach ensures that the student model benefits from the mentor's expertise without inheriting its mistakes. We provide theoretical analysis proving that MCD strictly generalizes vanilla KD and guarantees reduced negative transfer. We evaluate our Mentor-Critic Distillation across diverse teacher-student configurations on benchmark datasets, including CIFAR-100, ImageNet, and MedMNIST. Notably, MCD requires no architectural modifications or additional parameters, making it a practical drop-in replacement for standard knowledge distillation. These results highlight MCD's effectiveness in optimizing knowledge transfer and its robustness across diverse domains and data regimes, particularly in data-scarce scenarios typical of specialized domains such as medical imaging.

## 1 Introduction

The rapid advancements in deep neural networks (DNNs) (Krizhevsky et al., 2012; He et al., 2015; Vaswani et al., 2017; Liu et al., 2021) have revolutionized machine learning applications, powering tasks from image recognition to natural language processing with unprecedented accuracy. Yet, this progress comes with substantial challenges, as state-of-the-art models (Tan & Le, 2019; Dosovitskiy et al., 2020; Liu et al., 2021) often require considerable computational resources, making them impractical for deployment on resource-

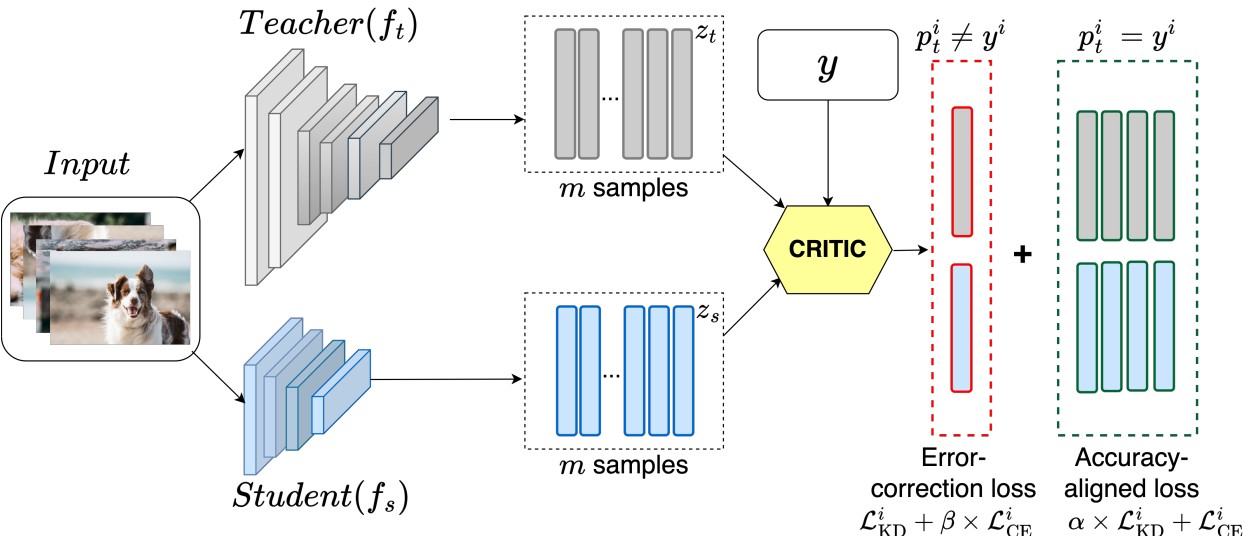

Figure 1: Overview of the Mentor-Critic Distillation (MCD) Framework. The input images are processed by both the teacher and student models to generate corresponding predictions. The Critic module then categorizes these predictions based on the correctness of the teacher's output. Correctly classified samples (green) are assigned higher weight for KL divergence loss, leveraging the teacher's reliable guidance to regularize the student. Incorrectly classified samples (red) emphasize cross-entropy loss, relying more on the ground truth to correct errors. This dynamic, correctness-based weighting mechanism enhances the regularization effect, improving the student model's generalization.

limited devices such as mobile phones and edge devices. Knowledge Distillation (KD) (Buciluă et al., 2006; Hinton et al., 2015) addresses this challenge by compressing large, high-performing models (known as teacher models) into more compact, efficient student models without substantial losses in predictive accuracy. KD transfers the dark knowledge from pre-trained teacher models to a compact and lightweight student model, making it an effective tool for deploying sophisticated models in real-world, constrained environments.

The classical KD approach (Hinton et al., 2015) transfers knowledge by minimizing the Kullback-Leibler (KL) divergence between the prediction logits of the teacher and student models. Although effective in its simplicity, this logit-based method has notable limitations (Tian et al., 2022; Yuan et al., 2019), as it considers only output logits, thereby missing out on valuable structural knowledge from the teacher model. In recent years, various KD techniques (Romero et al., 2015; Tian et al., 2022; Tung & Mori, 2019; Park et al., 2019; Zhao et al., 2022; Jin et al., 2023) have been developed to enhance the transfer of knowledge from teacher to student, broadly categorized into logit-based, feature-based, and relation-based methods.

Logit-based KD methods remain computationally efficient; however, they often fall short of capturing the complete depth of the teacher's knowledge, particularly in terms of inter-class relationships (Yuan et al., 2019). To address this, feature-based (Zagoruyko & Komodakis, 2017; Romero et al., 2015; Heo et al., 2019; Chen et al., 2021) and relation-based (Park et al., 2019; Tung & Mori, 2019) KD methods were introduced, leveraging intermediate feature representations and relationships among outputs, respectively, to provide richer information for the student model. Despite their performance benefits, these methods impose considerable computational and storage demands, which can limit their practicality in real-world applications (Heo et al., 2019). Recent efforts to improve KD have introduced diverse approaches, including Class Attention Transfer-Based Knowledge Distillation (CAT-KD) (Guo et al., 2023), an attention-based method however, it introduces additional computational overhead through changes in architecture, CAM (Selvaraju et al., 2017) extraction, and normalization.

Despite these advancements, a fundamental limitation persists: all prior methods apply *uniform* weighting regardless of teacher reliability. Consider a teacher model with 75% validation accuracy. Traditional KD applies the same distillation weight ($\lambda = 0.9$) to all samples—both the 75% where the teacher is re-

liable and the 25% where it errs. This creates two failure modes: (1) negative knowledge transfer from teacher mistakes, and (2) underutilized regularization from correct predictions. Recent methods like Decoupled Knowledge Distillation (DKD) (Zhao et al., 2022) decouple target/non-target components but remain correctness-agnostic, failing to address when the teacher itself is wrong. We hypothesize that dynamic, correctness-aware weighting can resolve both issues simultaneously.

Inspired by real-world learning—where students trust mentors when correct but consult external resources (like answer keys) when mentors err—we propose Mentor-Critic Distillation (MCD). MCD introduces correctness-aware weighting that dynamically adjusts knowledge transfer based on teacher reliability. Specifically, MCD employs two complementary mechanisms (Figure 1):

- **Accuracy-Aligned Knowledge Distillation (AAKD):** When the teacher is correct, amplify distillation loss ($\alpha \times \mathcal{L}_{\mathrm{KD}}$) to maximize regularization from reliable guidance.

- **Error-Correction Knowledge Distillation (ECKD):** When the teacher is incorrect, increase cross-entropy weight ($\beta \times \mathcal{L}_{\mathrm{CE}}$) to prioritize ground truth learning while still capturing inter-class relationships from the teacher's soft probabilities.

This simple modification requires no architectural changes or additional parameters—only a per-sample check of $\arg\max(p_t) = y$ to route training signals appropriately. Critically, this adaptive weighting achieves teacher influence configurations that no single fixed $\lambda$ can realize (Lemma 1), with formal guarantees that MCD attenuates negative transfer when the teacher errs (Corollary 1).

More importantly, our reformulation reveals that the classical KD loss is inherently a highly coupled formulation (as illustrated in Figure 1), which may explain the limitations of logit-based distillation. In Mentor-Critic Distillation (MCD), we identify two key issues stemming from this coupling. First, the cross-entropy (CE) loss term in traditional KD is insignificant and does not adapt based on the reliability of the teacher's predictions. This inflexibility leads to an inadequate emphasis on error correction when the teacher is incorrect, resulting in negative knowledge transfer. The inability to adjust the CE loss weight dynamically restricts the student model's capacity to learn effectively from the ground truth, especially in cases where the teacher model's guidance leads to an incorrect prediction. Second, the traditional formulation does not decouple the significance of knowledge transfer from correctly-predicted and incorrectly-predicted samples. In MCD, we propose treating the AAKD and ECKD losses independently. By decoupling these components, MCD allows for a more targeted and adaptive approach, where the AAKD loss benefits from the teacher's reliable and correct predictions, maximizing regularization and generalization. Conversely, ECKD emphasizes error correction when the teacher is incorrect, without suppressing the benefits of inter-class relationships provided by the teacher. This decoupling ensures that the contributions of both losses are optimized separately, leading to a more robust and effective distillation process.

Overall, our contributions are summarized as follows:

- **Correctness-aware distillation framework:** We introduce a simple yet effective approach that dynamically adjusts knowledge transfer based on teacher correctness. When the teacher is correct, we amplify distillation loss ($\alpha \times \mathcal{L}_{\mathrm{KD}}$) to maximize regularization; when incorrect, we emphasize cross-entropy ($\beta \times \mathcal{L}_{\mathrm{CE}}$) for error correction. This requires no architectural modifications or additional parameters.

- **Theoretical foundations:** We prove that MCD strictly generalizes vanilla KD (Lemma 1) and provide formal negative-transfer attenuation guarantees (Corollary 1).

- **Strong empirical validation:** Extensive experiments demonstrate consistent improvements over state-of-the-art methods: +3.6% over vanilla KD on CIFAR-100 (heterogeneous pairs), +1.6% on ImageNet, and +1.3% AUC on medical imaging tasks, with zero computational overhead. Ablation studies confirm that both AAKD and ECKD components are necessary for optimal performance.

## 2 Related Work

Knowledge Distillation (KD), introduced by Hinton et al. (2015), has become a cornerstone technique for model compression. Traditional KD methods align the output logits of teacher and student models using Kullback-Leibler (KL) divergence. Despite their simplicity, these methods often struggle to capture the full range of structural knowledge from the teacher, leading to limited student performance (Yuan et al., 2019).

Over the years, numerous KD techniques have been proposed, broadly categorized into *logit-based* (Cho & Hariharan, 2019; Furlanello et al., 2018; Mirzadeh et al., 2020; Zhao et al., 2022; Jin et al., 2023), *feature-based* (Romero et al., 2015; Tian et al., 2022; Chen et al., 2021; Heo et al., 2019), and *relation-based* (Park et al., 2019; Peng et al., 2019) methods. Feature-based approaches transfer intermediate representations but introduce significant computational overhead and storage costs (Heo et al., 2019). Relation-based methods distill inter-sample relationships but similarly increase memory requirements, limiting practical deployment.

Recent advancements have improved KD through decoupling strategies. Decoupled Knowledge Distillation (DKD) (Zhao et al., 2022) separates the KD loss into target and non-target components, allowing flexible weighting schemes. Multi-Level Logit Distillation (MLLD) (Jin et al., 2023) aligns predictions at instance, batch, and class levels but requires double the training epochs and heavy augmentation. Attention-based methods like CAT-KD (Guo et al., 2023) guide student focus using class activation maps but introduce architectural modifications. Despite these advances, all methods apply *uniform weighting regardless of teacher reliability*—none distinguishes samples where the teacher is correct from those where it errs.

Several works address teacher-student logit mismatch through normalization. Logit Standardization (LS-KD) (Sun et al., 2024) applies per-sample Z-score transformation to relax implicit magnitude-matching constraints imposed by shared temperature. LumiNet (Hossain et al., 2025) performs column-wise (per-class) standardization across the batch to capture intra-class dynamics. Both transformations are applied *uniformly to all samples*—they preserve relative logit structure but do not differentiate reliable from unreliable predictions. While they are orthogonal and combinable with MCD (see MCD+LS results), they do not address teacher misguidance, which our framework directly targets.

CRLD (Zhang et al., 2024) employs confidence-based binary filtering: predictions with maximum softmax probability below a threshold are excluded via Soft Label Selection. However, confidence serves only as a *proxy* for reliability—overconfident wrong predictions pass through while uncertain correct predictions are filtered out. CRLD does not verify whether the teacher's top prediction matches the ground truth. Wasserstein Knowledge Distillation (WKD) (Lv et al., 2024) replaces KL with Wasserstein distance using transport costs from Centered Kernel Alignment, while Scaled Decoupled Distillation (SDD) (Wei et al., 2024) decomposes logits into multi-scale local predictions. Both apply uniformly across samples without correctness-aware routing, and WKD incurs additional computational overhead from optimal transport.

Recent works explore automated distiller design. NORM (Liu et al., 2023) introduces N-to-One representation matching via expanded student features. DisWOT (Dong et al., 2023) enables training-free architecture search using zero-cost proxies. Auto-KD (Li et al., 2023b) and KD-Zero (Li et al., 2023a) employ Monte Carlo Tree Search and evolutionary algorithms to discover optimal configurations. These methods optimize *what* knowledge to transfer or *which* architecture to use, producing fixed recipes applied uniformly to all samples—they do not address *how much* to trust teacher predictions on a per-sample basis.

In contrast, MCD operates at a fundamentally different level: rather than modifying logits (calibration), discarding signal (thresholding), or searching configurations (AutoML), we dynamically reweight loss components based on an *explicit correctness signal*—checking whether the teacher's prediction matches the ground truth label. This preserves all teacher information, including valuable inter-class relationships from incorrect predictions, while controlling influence through instance-level gating. Our framework requires no architectural modifications, no additional parameters, and introduces negligible computational overhead, making it a practical drop-in replacement for standard KD.

## 3 Knowledge Distillation: A revisit

KD is a widely recognized model compression technique that leverages the knowledge of a large, pre-trained teacher model to train a smaller, more efficient student model. From a regularization standpoint, KD has

been shown to provide a form of implicit regularization (Yuan et al., 2020; Stanton et al., 2021; Ojha et al., 2023) that can improve the generalization capabilities (Zhou et al., 2021) of the student model . However, this regularization effect is often underexploited in traditional logit-based distillation frameworks.

We will explore this by revisiting the core mechanisms of KD from a regularization perspective, examining how the interaction between the distillation and ground truth losses influences the learning dynamics of the student model. Specifically, we investigate how the fixed weighting scheme between the distillation loss and cross-entropy loss limits the flexibility needed to adapt to variations in the teacher's reliability. By failing to independently adjust these weights based on the correctness of the teacher model, traditional KD approaches risk either overfitting to the teacher's incorrect knowledge or underutilizing the valuable regularization effect provided by the teacher's outputs.

*Notation:* Let $f_s$ be the student model and $f_t$ be the teacher model. This teacher model is pre-trained on the data $\mathcal{D}$. Let $z_t$ and $z_s$ represent the output (logits) of the teacher and student network, respectively. Let $p_t$ and $p_s$ represent the softmax outputs of teacher and student, respectively, and $\tau$ represents the temperature. Let $X$ represent the original data and $y$ represent its corresponding label. In general, $f$ refers to any deep learning model, $z$ refers to logits and $p$ refers to softmax probabilities.

In the traditional KD framework proposed by Hinton et al., the distillation loss is defined as:

$$\mathcal{L}_{\text{KD}} = \text{KL}\left(p^t(y|x,\tau) \,\|\, p^s(y|x,\tau)\right), \tag{1}$$

where $\text{KL}(\cdot \,\|\, \cdot)$ denotes the Kullback-Leibler (KL) divergence. Additionally, the student model is trained using the standard cross-entropy (CE) loss with the ground truth labels:

$$\mathcal{L}_{\text{CE}} = -\sum_y y_i \log p^s(y|x_i). \tag{2}$$

The total loss in traditional KD framework is a weighted combination of these two losses:

$$\mathcal{L}_{\text{total}} = \lambda \mathcal{L}_{\text{KD}} + (1-\lambda)\mathcal{L}_{\text{CE}}, \tag{3}$$

where $\lambda$ is a hyperparameter that controls the balance between the distillation loss and the CE loss.

Despite its widespread use, this setup has two limitations. First, the hyperparameter $\lambda$ is often set close to 0.9, ensuring that the distillation loss has a strong influence. However, this limits the potential to fully exploit the regularization benefits that come from the KD loss. Increasing the weight of the KD loss beyond 1.0 could provide substantial improvements in generalization due to smooth decision boundaries. In such cases, the amplified regularization effect aligns the student model more closely with the teacher's knowledge. Conversely, if the teacher is incorrect, a large weight on the distillation loss can result in negative knowledge transfer, reinforcing errors rather than correcting them. Traditional KD does not account for this variability in teacher reliability, thereby constraining the student model's overall effectiveness. Second, the relatively low weight assigned to the cross-entropy loss (typically around 0.1) limits the student's ability to learn directly from the ground truth, reducing its error-correction capability.

To understand this, we categorize the samples into two groups:

- **Correctly Classified Samples by the Teacher**: These are samples for which the teacher's prediction aligns with the ground truth $y_i$.

- **Incorrectly Classified Samples by the Teacher**: These are samples where the teacher's prediction does not match $y_i$.

The traditional KD can be reformulated as below:

$$\mathcal{L}_{\text{total}} = \frac{1}{N}\sum_{i=1}^{N}[\lambda \mathcal{L}_{\text{KD}} + (1-\lambda)\mathcal{L}_{\text{CE}}] * [\mathbb{I}_{correct}(x_i) + \mathbb{I}_{incorrect}(x_i)] \tag{4}$$

where,

$$\mathbb{I}_{\text{correct}}(x_i) = \begin{cases} 1 & \text{if } \arg\max(p^t(y|x_i)) = y_i, \\ 0 & \text{otherwise,} \end{cases}$$

$$\mathbb{I}_{\text{incorrect}}(x_i) = 1 - \mathbb{I}_{\text{correct}}(x_i).$$

After reformulating the traditional KD loss, we can analyze its behavior under two key scenarios: when the teacher model is incorrect and when it is correct. These scenarios highlight the potential pitfalls of the fixed-weight approach in traditional KD frameworks.

**Case 1: Teacher Model is Incorrect**: When the teacher model makes an incorrect prediction, the term $\lambda \times \mathcal{L}_{KD}$ still exerts a significant influence on the student model's learning process. Since $\lambda$ is typically set to a high value, such as 0.9, the distillation loss drives the student to mimic the teacher's erroneous outputs, resulting in negative knowledge transfer. In this case, the cross-entropy loss $(1 - \lambda) \times \mathcal{L}_{CE}$, which should help correct the student's learning by reinforcing the ground truth, has minimal impact because of its small weight (0.1). As a result, the student model is inadequately equipped to override the teacher's incorrect guidance, limiting its ability to learn the true data distribution effectively.

**Case 2: Teacher Model is Correct**: When the teacher model provides correct predictions, the distillation term $\lambda \times \mathcal{L}_{KD}$ does facilitate beneficial knowledge transfer from the teacher to the student. However, because $\lambda$ is capped at 0.9, the student model is restricted in learning from the teacher's knowledge. This cap constrains the potential for the student to fully leverage the teacher's correct and informative outputs.

**Can We Simply Scale $\mathcal{L}_{KD}$ to a Higher Value?** One might wonder whether we can address this limitation by simply increasing the weight of the distillation loss $\lambda$ beyond 0.9 to enhance knowledge transfer from the teacher. However, scaling $\mathcal{L}_{KD}$ to a higher value introduces a risk: it amplifies the negative transfer of knowledge when the teacher model is incorrect. This trade-off highlights the inherent challenge in traditional KD's fixed weighting mechanism limiting it to suboptimal performance.

These observations emphasize the necessity of a more balanced and adaptive approach to knowledge distillation.

## 4 Mentor Critic Distillation

To address the limitations of traditional KD, we propose the Mentor-Critic Distillation (MCD) framework. MCD introduces an adaptive and dynamic approach to balancing the distillation and cross-entropy losses based on the teacher model's reliability. The traditional KD loss formulation can be expressed as:

$$\mathcal{L}_{\text{total}} = \frac{1}{N} \sum_{i=1}^{N} \left[ \mathbb{I}_{correct}(x_i) * \left[ \lambda \mathcal{L}_{\text{KD}}^i + (1-\lambda)\mathcal{L}_{\text{CE}}^i \right] + \mathbb{I}_{incorrect}(x_i) * \left[ \lambda \mathcal{L}_{\text{KD}}^i + (1-\lambda)\mathcal{L}_{\text{CE}}^i \right] \right] \qquad (5)$$

We redefine this loss in the MCD framework to incorporate dynamic weighting:

$$\mathcal{L}_{\text{MCD}} = \frac{1}{N} \sum_{i=1}^{N} \left[ \underbrace{\mathbb{I}_{\text{correct}}(x_i) \times \left( \alpha \times \mathcal{L}_{\text{KD}}^i + \mathcal{L}_{\text{CE}}^i \right)}_{\text{Accuracy-Aligned Loss}} + \underbrace{\mathbb{I}_{\text{incorrect}}(x_i) \times \left( \mathcal{L}_{\text{KD}}^i + \beta \times \mathcal{L}_{\text{CE}}^i \right)}_{\text{Error-Correction Loss}} \right] \qquad (6)$$

$$\mathcal{L}_{\text{MCD}} = \mathcal{L}_{\text{AAKD}} + \mathcal{L}_{\text{ECKD}} \qquad (7)$$

where $\alpha$ and $\beta$ are hyper-parameters that adjust the influence of the distillation and cross-entropy losses based on the teacher's correctness. $\mathcal{L}_{\text{AAKD}}$ is the loss component aligned with samples where the teacher is correct and $\mathcal{L}_{\text{ECKD}}$ is the loss component that handles samples where the teacher is incorrect. Implementation-oriented pseudocode is shown in **Appendix A.1**.

A student relies on a mentor for guidance but also consults reliable resources when the mentor provides incorrect information. Similarly, the student model in MCD dynamically adjusts its learning strategy. When

the teacher is correct, MCD emphasizes knowledge transfer from the teacher, using a higher weight $\alpha$ to maximize learning. Conversely, when the teacher is incorrect, MCD reduces the influence of the teacher and increases the weight $\beta$ on the cross-entropy loss, allowing the student to rely more heavily on the ground truth.

**Can we discard $\mathcal{L}_{CE}$ in $\mathcal{L}_{AAKD}$ and $\mathcal{L}_{KD}$ in $\mathcal{L}_{ECKD}$?**
The $\mathcal{L}_{CE}$ term in $\mathcal{L}_{AAKD}$ ensures that the student model continues to learn directly from the ground truth, even when the teacher model provides over-generalized, though correct, predictions. On the other hand, the $\mathcal{L}_{KD}$ term in $\mathcal{L}_{ECKD}$ remains valuable even when the teacher makes mistakes. The softened probability distribution from the teacher still contains critical information about inter-class relationships and similarities. By including $\mathcal{L}_{KD}$ in $\mathcal{L}_{ECKD}$, the student model can leverage the teacher's overall understanding of the problem space, resulting in smoother decision boundaries and improved generalization.

## 5   Theoretical Analysis

**Setup and notation.**   Let $p_s(\cdot \mid x)$ and $p_t(\cdot \mid x)$ denote the student and teacher softmax distributions, and let $y$ be the one-hot label. We omit the temperature parameter $\tau$ for clarity. The per-logit gradients used below are the standard identities: $\nabla_z \mathrm{CE}(y, p_s) = p_s - y$ and $\nabla_z \mathrm{KL}(p_t \| p_s) = p_s - p_t$.

**Proposition 1** (Mixture-target view of vanilla KD)**.**  *Consider vanilla KD with a fixed $\lambda \in (0, 1)$:*

$$L_{KD} = \lambda \, \mathrm{KL}(p_t \| p_s) + (1 - \lambda) \, \mathrm{CE}(y, p_s).$$

*Then*

$$\nabla_z L_{KD} = \lambda(p_s - p_t) + (1 - \lambda)(p_s - y) = p_s - \big((1 - \lambda)\, y + \lambda \, p_t\big),$$

*which shows that vanilla KD is equivalent (up to the usual positive scalar factors) to ERM against the sample-independent "soft target" $\tilde{y} = (1 - \lambda)\, y + \lambda \, p_t$ for every training point.*

**Proposition 2** (Instance-dependent mixture target in MCD)**.**  *Let $\alpha, \beta > 0$ be the MCD weights. For a sample with a* correct *teacher prediction, MCD uses $L_{corr} = \alpha \, \mathrm{KL}(p_t \| p_s) + \mathrm{CE}(y, p_s)$, giving*

$$\nabla_z L_{corr} = \alpha(p_s - p_t) + (p_s - y) = (\alpha + 1)\left[ p_s - \frac{\alpha \, p_t + y}{\alpha + 1} \right].$$

*For an* incorrect *teacher prediction, MCD uses $L_{inc} = \mathrm{KL}(p_t \| p_s) + \beta \, \mathrm{CE}(y, p_s)$, giving*

$$\nabla_z L_{inc} = (p_s - p_t) + \beta(p_s - y) = (\beta + 1)\left[ p_s - \frac{p_t + \beta \, y}{\beta + 1} \right].$$

*Hence MCD performs ERM against an* instance-dependent *target*

$$\tilde{y}(x) = \begin{cases} \dfrac{\alpha \, p_t + y}{\alpha + 1}, & \text{if the teacher is correct,} \\[2mm] \dfrac{p_t + \beta \, y}{\beta + 1}, & \text{if the teacher is incorrect.} \end{cases}$$

**Lemma 1** (No-single-$\lambda$ lemma (strict expressivity gap))**.**  *Let $w_{corr} = \alpha/(\alpha + 1)$ and $w_{inc} = 1/(\beta + 1)$ be the effective teacher weights on correct and incorrect samples under MCD. For any $\lambda_0 \in (0, 1)$, there exist $(\alpha, \beta)$ such that $w_{corr} > \lambda_0$ and $w_{inc} < \lambda_0$ simultaneously. No scalar $\lambda$ in vanilla KD can realize both inequalities at once, because vanilla KD fixes a* single *teacher weight $\lambda$ for all samples (correct and incorrect).*

*Proof.* Choose $\alpha > \lambda_0/(1 - \lambda_0)$ so that $w_{\mathrm{corr}} = \alpha/(\alpha + 1) > \lambda_0$, and $\beta > 1/\lambda_0 - 1$ so that $w_{\mathrm{inc}} = 1/(\beta + 1) < \lambda_0$. Vanilla KD has $w_{\mathrm{corr}} = w_{\mathrm{inc}} = \lambda$ by construction, hence cannot satisfy $w_{\mathrm{corr}} > \lambda_0$ and $w_{\mathrm{inc}} < \lambda_0$ simultaneously. $\qquad \square$

**Corollary 1** (Negative-transfer attenuation guarantee)**.**  *Consider a sample where the teacher's top class is wrong. Under vanilla KD, the teacher distribution enters the target with weight $\lambda$. Under MCD, the teacher weight on such samples is $w_{inc} = 1/(\beta + 1)$. For any fixed $\lambda \in (0, 1)$, choosing $\beta > \frac{1}{\lambda} - 1$ guarantees $w_{inc} < \lambda$, i.e., the wrong-class mass carried from $p_t$ into the training signal is strictly smaller under MCD than under vanilla KD.*

*Remark* (Recoverability and strict generalization). Setting $\alpha = \lambda/(1-\lambda)$ and $\beta = 1/\lambda - 1$ yields $w_{\text{corr}} = w_{\text{inc}} = \lambda$; if we ignore correctness gating, MCD reduces to vanilla KD. Conversely, vanilla KD cannot realize $w_{\text{corr}} \neq w_{\text{inc}}$, so MCD strictly generalizes vanilla KD in the class of per-sample targets. Moreover, in contrast to DKD—which decouples target/non-target components but is correctness-agnostic—MCD's gating by teacher correctness yields the attenuation guarantee above.

**Interpretation** MCD *selectively* increases the regularizing KL term when the teacher is trustworthy (large $\alpha$ on correct samples) and down-weights it when the teacher is unreliable (small $1/(\beta + 1)$ on incorrect samples), while compensating with CE. This correctness-aware control is unavailable to correctness-agnostic formulations (vanilla KD, DKD) and explains the empirical gains observed in our experiments.

## 6 Experiments

Our experimental study is designed to test the central premise of MCD—trust the teacher when it is correct, defer to ground truth when it is not—across diverse architectures and data regimes. We first establish accuracy gains on CIFAR-100 and ImageNet with both homogeneous and heterogeneous pairs, then examine data-scarce medical imaging tasks. Finally, we analyze sensitivity to $(\alpha, \beta)$ and verify that MCD introduces no training-time overhead.

Datasets and protocols are summarized in Appendix B (with MedMNIST in Appendix B.1); pseudocode and training setup appear in Appendix A.1–A.2. We report Top-1 for classification benchmarks and AUC for clinical datasets.

Table 1: Experiment results on CIFAR100 dataset for heterogeneous teacher-student pairs. The $\Delta$ represents the performance gain over the traditional KD. All the results are averaged over 3 trials. The best results are highlighted in bold. Eff. Iter: effective training iterations relative to standard 240-epoch training (MLLD uses 480 epochs and 2× forward passes). Here vKD+LS is vanilla KD with LS (Sun et al., 2024).

| Distillation type | Eff. Iter | Teacher
Student | ResNet32x4
79.42
SV1
70.50 | WRN40-2
75.61
SV1
70.50 | VGG13
74.64
MV2
64.60 | ResNet50
79.34
MV2
64.60 | ResNet32x4
79.42
SV2
71.82 | Avg |
|---|---|---|---|---|---|---|---|---|
| feature | 1× | FitNet (Romero et al., 2015) | 73.59 | 73.73 | 64.14 | 63.16 | 73.54 | 69.63 |
| | | RKD (Park et al., 2019) | 72.28 | 72.21 | 64.52 | 64.43 | 73.21 | 69.33 |
| | | CRD (Tian et al., 2022) | 75.11 | 76.05 | 69.73 | 69.11 | 75.65 | 73.13 |
| | | OFD (Heo et al., 2019) | 75.98 | 75.85 | 69.48 | 69.04 | 76.82 | 73.43 |
| | | ReviewKD (Chen et al., 2021) | **77.45** | 77.14 | **70.37** | 69.89 | **77.78** | 74.53 |
| logit | 1× | KD (Hinton et al., 2015) | 74.07 | 74.83 | 67.37 | 67.35 | 74.45 | 71.60 |
| | | CTKD (Li et al., 2023c) | 74.48 | 75.78 | 68.46 | 68.47 | 75.31 | 72.5 |
| | | DKD (Zhao et al., 2022) | 76.45 | 76.70 | 69.71 | 70.35 | 77.07 | 74.05 |
| | | MCD | 77.41 | **77.38** | 70.11 | **71.08** | 77.35 | **74.66** |
| | | $\Delta$ | +3.34 | +2.55 | +2.74 | +3.73 | +2.9 | +3.6 |
| logit | 4× | MLLD(Jin et al., 2023) | 77.18 | 77.44 | 70.57 | 71.04 | 78.44 | 74.93 |
| | | MCD+MLLD | **77.52** | **77.68** | **70.71** | **71.19** | **79.03** | **75.22** |
| logit+LS | 1× | vKD+LS | 74.27 | 75.38 | 68.43 | 68.75 | 75.34 | 72.43 |
| | | KD+LS(Sun et al., 2024) | 74.44 | 75.64 | 68.61 | 69.02 | 75.56 | 72.65 |
| | | MCD+LS | **77.45** | **77.42** | **70.41** | **71.21** | **77.39** | **74.77** |

### 6.1 CIFAR-100

CIFAR-100 serves as a deliberate stress test: teacher models exhibit severe overfitting (>99% training vs. ~75% validation accuracy, Table 8), minimizing ECKD activation and creating conditions least favorable to MCD. We retain teachers in training mode to introduce stochasticity via dropout and batchnorm. Despite this least-favorable setting, MCD achieves +3.48% improvement over vanilla KD (Table 2), demonstrating that correctness-aware gating provides consistent benefits even when teacher errors are rare. Appendix D.3 provides a detailed component analysis isolating the contribution of each branch.

Table 2: Experiment results on CIFAR100 dataset for homogeneous teacher-student pairs. The $\Delta$ represents the performance gain over the traditional KD. All the results are averaged over 3 trials. The best results are highlighted in bold. Eff. Iter: effective training iterations relative to standard 240-epoch training (MLLD uses 480 epochs and 2× forward passes). Here vKD+LS is vanilla KD with LS (Sun et al., 2024).

| Distillation type | Eff. Iter | Teacher / Student | ResNet56 72.34 ResNet20 69.06 | ResNet110 74.31 ResNet32 71.14 | ResNet32x4 79.42 ResNet8x4 72.50 | WRN40-2 75.61 WRN16-2 73.26 | WRN40-2 75.61 WRN40-1 71.98 | VGG13 74.64 VGG8 70.36 | Avg |
|---|---|---|---|---|---|---|---|---|---|
| feature | 1× | FitNet (Romero et al., 2015) | 69.21 | 71.06 | 73.50 | 73.58 | 72.24 | 71.02 | 71.77 |
| | | RKD (Park et al., 2019) | 69.61 | 71.82 | 71.90 | 73.35 | 72.22 | 71.48 | 71.73 |
| | | CRD (Tian et al., 2022) | 71.16 | 73.48 | 75.51 | 75.48 | 74.14 | 73.94 | 73.95 |
| | | OFD (Heo et al., 2019) | 70.98 | 73.23 | 74.95 | 75.24 | 74.33 | 73.95 | 73.78 |
| | | ReviewKD (Chen et al., 2021) | 71.89 | 73.89 | 75.63 | 76.12 | 75.09 | 74.84 | 74.58 |
| logit | 1× | KD (Hinton et al., 2015) | 70.66 | 73.08 | 73.33 | 74.92 | 73.54 | 72.98 | 73.09 |
| | | CTKD (Li et al., 2023c) | 71.19 | 73.52 | 73.39 | 75.45 | 73.93 | 73.52 | 73.5 |
| | | DKD (Zhao et al., 2022) | **71.97** | **74.11** | 76.32 | **76.24** | 74.81 | 74.68 | 74.68 |
| | | **MCD** | 71.83 | 74.10 | **76.81** | 76.07 | **75.11** | **74.97** | **74.81** |
| | | $\Delta$ | +1.17 | +1.02 | +3.48 | +1.15 | +1.57 | +1.99 | +1.72 |
| logit | ≥4× | MLLD(Jin et al., 2023) | 72.19 | 74.11 | 77.08 | 76.63 | 75.35 | 75.18 | 75.09 |
| | | MCD+MLLD | **72.25** | **74.47** | **77.45** | **76.71** | **75.52** | **75.24** | **75.27** |
| logit+LS | 1× | vKD+LS | 70.15 | 72.56 | 74.04 | 74.91 | 73.58 | 73.84 | 73.18 |
| | | KD+LS(Sun et al., 2024) | 71.43 | 74.17 | 76.62 | 76.11 | 74.37 | 74.36 | 74.51 |
| | | MCD+LS | **71.98** | **74.27** | **77.11** | **76.22** | **75.28** | **75.11** | **74.99** |

Table 3: Cross-architecture distillation on CIFAR-100. We evaluate MCD on ViT→CNN and CNN→ViT configurations. $\Delta$: MCD improvement over vanilla KD. Training accuracy are reported in Table 8

| Teacher | Student | T. Acc | S. Acc | feature | | | logit | | | |
|---|---|---|---|---|---|---|---|---|---|---|
| | | | | FitNet | RKD | CRD | KD | DKD | **MCD** | $\Delta$ |
| Swin-T | ResNet18 | 89.26 | 74.01 | 78.87 | 74.11 | 77.63 | 78.74 | 80.26 | **81.74** | +3.00 |
| ConvNeXt-T | DeiT-T | 88.41 | 68.00 | 60.78 | 69.79 | 65.94 | 72.99 | 74.60 | **77.21** | +4.22 |

Despite these limitations, MCD achieves superior performance across various teacher-student pairs compared to existing baselines. The previous state-of-the-art method (under the standard augmentation and 240-epoch training configuration), DKD, decouples distillation loss into Target Class (TCKD) and Non-Target Class Knowledge Distillation (NCKD), with NCKD proving particularly effective. DKD implicitly increases distillation influence by raising NCKD weights up to 8.0 for CIFAR-100 (where teachers overfit) but drops to 0.5 for ImageNet (where teachers are well-calibrated)—inadvertently supporting our hypothesis that distillation weights should adapt to teacher reliability. Similarly, MLLD employs multi-level alignment (batch, class, instance) with prediction augmentation across multiple temperatures, effectively amplifying distillation loss through implicit weight scaling. MLLD has overall best CIFAR-100 performance but uses extended training configuration with 2 × forward passes (weak + strong data augmentation).

While LS-KD is orthogonal and combinable with MCD (see MCD+LS results), it does not address teacher misguidance—our framework's core focus. Tables 1 and 2 demonstrate that MCD+LS consistently outperforms KD+LS, confirming correctness-aware supervision's pivotal role over global logit normalization. Our approach achieves comparable or superior results through explicit, dynamic weighting based on teacher correctness, without requiring complex multi-level alignment, architectural modifications, or heavy data augmentation strategies employed by MLLD. These results underscore that correctness-sensitive guidance is more critical than alignment complexity for effective student training. Our approach shows that such complex alignment mechanisms are not always necessary. Dynamically increasing the weight of the KL divergence loss based on the teacher correctness efficiently boosts the student performance. Runtime comparison supporting 'no overhead' is in Appendix D (Figure 2.)

**Cross-Architecture Distillation with Vision Transformers:** To evaluate MCD's generalization beyond CNNs, we test two challenging cross-architecture settings: ViT→CNN (Swin-Tiny→ResNet18) and CNN→ViT (ConvNeXt-Tiny→DeiT-Tiny). Table 3 shows MCD achieves +3.00% and +4.22% over vanilla

KD respectively—substantially larger gains than CNN-only experiments. These teachers exhibit natural error rates (5–7%, Table 8) compared to overfitted CNN teachers, providing more ECKD activation.

**Impact of Image Augmentation with Temperature Regularization:** To investigate the effects of incorporating multiple temperatures as a regularization strategy in our MCD framework, we experimented with MLLD-similar framework by applying multiple temperatures, a combination of strong and weak augmentation. This is because MLLD has an advantage of $2\times$ augmented datapoints, 2 to $4\times$ epochs, and multiple temperatures as compared to other baselines such as MLLD (Jin-Ying). We observe that varying temperatures introduced diverse levels of regularization, which enhanced the model's performance. The results are shown in Table 1 and 2 as MCD+MLLD. The core concept revolves around effectively transferring both the quality and magnitude of regularization to the student model, ensuring that the student benefits from a balanced and robust learning process.

In Table 1 and 2, vKD+LS, represents vanilla KD with LS and KD+LS is weighted KD as described in Sun et al. (2024). All SOTA distillation methods work by scaling up the knowledge transfer from teacher to student either implicitly (MLLD adds 3 losses each with 4 different temperatures and puts weight on them) or explicitly (e.g., DKD puts 8.0 weight on the loss component, LS (KD+LS) puts weight of 9.0 on KL divergence loss); MCD's novelty lies in its simplicity and decoupling strategy for maximum performance. Further, every distillation method uses its own temperature that suits their strategy.

**Impact of Homogeneous vs. Heterogeneous Teacher-Student Pairs.** The effectiveness of MCD varies based on structural similarity between teacher and student. In both settings, CIFAR-100 teachers achieve very high training accuracy ($>97\%$, Table 8), meaning the AAKD branch dominates while ECKD receives limited activation. However, heterogeneous pairs show larger gains ($+3.6\%$ average) compared to homogeneous pairs ($+1.72\%$ average). We attribute this to the larger *representation gap* in heterogeneous settings: when teacher and student architectures differ substantially (e.g., ResNet32$\times$4 $\rightarrow$ ShuffleNet-V1), the student benefits more from AAKD's distillation ($\alpha \times \mathcal{L}_{\text{KD}}$), which provides stronger regularization to bridge architectural differences. In homogeneous settings, where teacher and student share similar inductive biases, vanilla KD already transfers knowledge effectively, leaving less room for amplification. Additionally, heterogeneous students often start with lower baseline accuracy (e.g., MobileNetV2 at 64.60% vs. ResNet8x4 at 72.50%), providing more room for improvement.

We note that MLLD performs strongly due to its use of heavy data augmentation ($2\times$ input views, $\geq 2\times$ training). However, MCD+MLLD exceeds MLLD alone, indicating complementarity. Across both configurations, MCD consistently outperforms KD+LS, confirming its strength as a base distillation strategy.

## 6.2 ImageNet

The teacher models trained on ImageNet are typically well-calibrated (Guo et al., 2017), with minimal differences between training and validation accuracies Table 8, indicating strong generalization capabilities. In our experiments, we evaluate two distinct teacher-student pairs: a homogeneous pair (ResNet34 - ResNet18), and a heterogeneous pair (ResNet50 - MobileNetV1). The results are shown in Table 4. Our approach shows significant performance improvement over traditional KD while also outperforming existing baselines, achieving these results without relying on prediction augmentation or introducing additional parameters. Notably, MCD outperforms recent methods including LumiNet (Hossain et al., 2025), WKD-L (Lv et al., 2024), and CRLD (Zhang et al., 2024) on the heterogeneous pair, despite these methods requiring additional computational overhead (offline cost matrices or $2\times$ forward passes). It is noteworthy that, unlike MLLD (Jin et al., 2023), which relies on a combination of weak and strong data augmentations to enhance performance, our approach does not require such augmentations. Despite this, our method still achieves better results compared to MLLD. To disentangle the roles of AAKD and ECKD, we include an isolation study in Appendix D.2 (Table 9).

## 6.3 Impact of $\alpha$ and $\beta$

MCD requires balancing two competing objectives, which we term the *Knowledge Transfer Trade-off*: avoiding both overfitting to ground truth labels and over-reliance on teacher guidance. Table 5 reveals this critical equilibrium.

Table 4: Results on ImageNet dataset. We report Top-1/Top-5 accuracy for one homogeneous (ResNet34→ResNet18) and one heterogeneous (ResNet50→MobileNetV1) pair. All results averaged over 3 trials. Overhead: training time relative to vanilla KD. †Requires offline cost matrix computation. ‡Uses 2× forward passes(RandAugment + Cutout).

| Distillation type | Overhead | Method | Top-1 | Top-5 | Top-1 | Top-5 |
|---|---|---|---|---|---|---|
| | | | ResNet34→ResNet18 | | ResNet50→MobileNetV1 | |
| | | Teacher | 73.31 | 91.42 | 76.16 | 92.86 |
| | | Student | 69.75 | 89.07 | 68.87 | 88.76 |
| feature | 1.2–1.5× | AT | 70.69 | 90.01 | 69.56 | 89.33 |
| | | OFD (Heo et al., 2019) | 70.81 | 89.98 | 71.25 | 90.34 |
| | | CRD (Tian et al., 2022) | 71.17 | 90.13 | 71.37 | 90.41 |
| | | ReviewKD (Chen et al., 2021) | 71.61 | 90.51 | 72.56 | 91.00 |
| attention | 1.1× | CAT-KD (Guo et al., 2023) | 71.26 | 90.45 | 72.24 | 91.13 |
| logit | 1.0× | KD (Hinton et al., 2015) | 70.66 | 89.88 | 68.58 | 88.98 |
| | | TAKD (Mirzadeh et al., 2020) | 70.78 | 90.16 | 70.82 | 90.01 |
| | | DKD (Zhao et al., 2022) | 71.70 | 90.41 | 72.05 | 91.05 |
| | | KD+LS (Sun et al., 2024) | 71.42 | 90.29 | 72.18 | 90.80 |
| | | SD-KD (Wei et al., 2024) | 71.44 | 90.05 | 72.24 | 90.71 |
| | | LumiNet (Hossain et al., 2025) | 72.16 | 90.60 | 72.55 | 91.12 |
| | | **MCD (Ours)** | 72.08 | 90.59 | **73.65** | **91.88** |
| | 1.3×† | WKD-L (Lv et al., 2024) | **72.49** | **90.75** | 73.17 | 91.32 |
| | 1.75×‡ | CRLD (Zhang et al., 2024) | 72.05 | 90.74 | 73.15 | 91.54 |
| | ≥ 2.0× | MLLD (Jin et al., 2023) | 71.90 | 90.41 | 73.01 | 91.42 |

Table 5: Impact of hyperparameters $\alpha$ and $\beta$ on CIFAR-100. Left: varying $\alpha$ with $\beta = 2$ fixed. Right: varying $\beta$ with $\alpha = 12$ fixed. Best results in **bold**.

| Teacher | Student | $\beta = 2$, varying $\alpha$ | | | | $\alpha = 12$, varying $\beta$ | | | |
|---|---|---|---|---|---|---|---|---|---|
| | | 1 | 8 | 12 | 16 | 1 | 2 | 4 | 6 |
| ResNet50 | MobileNet-V2 | 67.57 | **71.22** | 71.07 | 70.13 | 70.61 | **71.07** | 70.81 | 70.64 |
| ResNet32x4 | ShuffleNet-V2 | 74.52 | 77.50 | **77.71** | 77.44 | 77.23 | **77.71** | 77.44 | 77.34 |
| WRN-40-2 | WRN-16-2 | 74.97 | **76.15** | 76.08 | 75.71 | 75.80 | **76.08** | 75.75 | 75.65 |

**Excessive CE weighting (high $\beta$):** When the CE loss dominates, the student overfits to training labels, achieving high training accuracy but poor generalization on validation data. The CE loss encourages sharp, high-confidence predictions aligned with ground truth, causing the student to memorize training examples while missing the inter-class relationships encoded in the teacher's soft probabilities. This is analogous to rote memorization without conceptual understanding.

**Excessive KL weighting (high $\alpha$):** Conversely, overemphasizing the KL divergence loss causes the student to blindly mimic the teacher's outputs, including its biases and limitations. While KL-based regularization benefits generalization, excessive weight prevents the student from learning task-specific details from ground truth labels, degrading performance on precise discrimination tasks. The student becomes proficient in the teacher's generalized concepts but lacks accuracy on specific cases.

**Optimal balance:** MCD addresses this trade-off through correctness-aware weighting. When the teacher is correct ($\alpha \times \mathcal{L}_{\mathrm{KD}}$), we amplify regularization to leverage reliable guidance. When incorrect ($\beta \times \mathcal{L}_{\mathrm{CE}}$), we increase supervision from ground truth while retaining inter-class relationships from teacher probabilities. Our default values ($\alpha = 12, \beta = 2$) achieve robust performance across diverse teacher-student configurations, demonstrating that dynamic, correctness-based weighting effectively navigates the overfitting-overregularization spectrum.

Table 6: Diagnostic performance on medical imaging tasks. Results shown as AUC/ACC. Best student performance in **bold**.

| Method | Teacher | Student | DermaMNIST | | BreastMNIST | | OrganMNIST-S | |
|---|---|---|---|---|---|---|---|---|
| | | | AUC | ACC | AUC | ACC | AUC | ACC |
| *Teacher* | ResNet-50 | – | 0.9484 | 82.59 | 0.8726 | 82.69 | 0.9794 | 82.60 |
| *Student-only* | – | SV2 | 0.8755 | 72.72 | 0.8099 | 79.49 | 0.9734 | 76.79 |
| *Student-only* | – | MV2 | 0.9040 | 75.31 | 0.8181 | 82.05 | 0.9776 | 79.76 |
| KD (Hinton et al., 2015) | ResNet-50 | SV2 | 0.9012 | 73.82 | 0.8348 | 83.33 | 0.9778 | 80.71 |
| KD (Hinton et al., 2015) | ResNet-50 | MV2 | 0.9075 | 75.51 | 0.8617 | 84.62 | 0.9796 | 81.96 |
| DKD (Zhao et al., 2022) | ResNet-50 | SV2 | 0.9089 | 74.66 | 0.8823 | 84.62 | 0.9789 | 81.70 |
| DKD (Zhao et al., 2022) | ResNet-50 | MV2 | 0.9121 | 76.41 | 0.8742 | 85.90 | 0.9808 | 82.80 |
| **MCD** | ResNet-50 | SV2 | **0.9119** | **76.51** | **0.8995** | **85.26** | **0.9797** | **82.58** |
| **MCD** | ResNet-50 | MV2 | **0.9205** | **78.70** | **0.8839** | **86.54** | **0.9822** | **83.55** |

## 6.4 Evaluation on Medical Imaging Tasks

To assess real-world applicability, we evaluate MCD on medical imaging tasks spanning dermatoscopic, ultrasound, and CT modalities using MedMNIST v2 (Yang et al., 2023) (dataset specs and training details are in Appendix B.1). These settings are data-scarce, exhibit high inter-class similarity, and demand stringent accuracy for decision support.

**Results:** Table 6 presents the diagnostic performance across all methods. We report Area Under the ROC Curve (AUC) and Accuracy (ACC) following medical imaging evaluation standards. MCD consistently outperforms traditional KD and state-of-the-art DKD across all three medical imaging datasets. MobileNetV2 with MCD achieves improvements of +1.30% AUC on DermaMNIST, +2.22% AUC on BreastMNIST, and +0.26% AUC on OrganMNIST-S over baseline KD, demonstrating effective knowledge transfer across diverse medical imaging modalities. These results demonstrate that MCD effectively addresses a key challenge in medical AI: compressing accurate diagnostic models for resource-constrained clinical devices while maintaining diagnostic performance. The correctness-aware distillation mechanism is particularly valuable in medical imaging where inheriting teacher errors could have serious clinical consequences.

## 7 Limitations and Future Directions

MCD has two primary limitations. First, it uses a **binary correctness signal** ($\arg\max(p_t) = y$), treating all correct predictions equally regardless of confidence—a confident prediction ($p_t = 0.95$) receives the same weight as a barely correct one ($p_t = 0.35$). Second, **hyperparameters** $\alpha$ and $\beta$ require manual tuning, though our defaults ($\alpha = 12$, $\beta = 2$) generalize well across diverse teacher-student configurations.

Two natural extensions address these limitations. **Confidence-aware soft weighting** could modulate $\alpha$ and $\beta$ based on prediction confidence, enabling smoother adaptation that leverages teacher calibration beyond binary correctness. **Automatic hyperparameter selection** via meta-learning could infer optimal values from teacher properties (validation accuracy, calibration error), eliminating manual tuning and enhancing practical deployability across domains.

## 8 Conclusion

We introduced Mentor-Critic Distillation (MCD), a correctness-aware framework that dynamically adjusts knowledge transfer based on teacher reliability through two complementary mechanisms: Accuracy-Aligned KD ($\alpha \times \mathcal{L}_{\mathrm{KD}}$) for correct predictions and Error-Correction KD ($\beta \times \mathcal{L}_{\mathrm{CE}}$) for incorrect ones. We prove that MCD strictly generalizes vanilla KD (Lemma 1) with formal negative transfer attenuation guaran-

tees (Corollary 1), while requiring zero architectural modifications or additional parameters. Extensive experiments demonstrate consistent improvements, with ablations confirming both components are essential. MCD provides a theoretically grounded, empirically validated solution that explicitly accounts for teacher imperfection—a critical consideration for real-world deployment. Future work includes confidence-aware soft weighting and automatic hyperparameter selection to further enhance practical deployability.

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

# A    Pseudocode and Implementation Details

## A.1    PyTorch-style Pseudocode

---
**Algorithm 1:** PyTorch-style pseudocode for Mentor-Critic Distillation
---

```
# Inputs:
# z_s:  Logits from student
# z_t:  Logits from teacher
# y:  Ground truth
# T: Temperature
# alpha, beta:  hyper-parameters for MCD
# Compute correct/incorrect masks for teacher's predictions
mask_c = (argmax(z_t, dim=1) == y) # Correct predictions
mask_ic = not mask_c # Incorrect predictions
p_s = F.softmax(z_s / T, dim=1)
p_t = F.softmax(z_t / T, dim=1)
# Compute Accuracy Aligned loss
aa_kd = alpha * KLD(p_s[mask_c], p_t[mask_c]) +
        CE(z_s[mask_c], y[mask_c])
# Compute Error Correction loss
ec_kd = KLD(p_s[mask_ic], p_t[mask_ic]) +
beta * CE(z_s[mask_ic], y[mask_ic])
# Compute total MCD loss
mcd_loss = aa_kd + ec_kd
# Return the MCD loss
return mcd_loss
```

---

## A.2    Experiment Setup and details

We adopt the same experimental settings as in previous studies (Zhao et al., 2022; Jin et al., 2023; Sun et al., 2024) for CIFAR-100 and ImageNet. All results are averaged over three independent trials. The optimizer is SGD and the learning rate is multiplied by a factor of 0.1 after $[150, 180, 210]$ epochs. The total number of epochs is 240, except for MLLD and MCD+MLLD, where the number of epochs is 480. By default, temperature is set to 20, $\beta$ is set to 2.0 and $\alpha$ is set to 12. For CNN-ViT and ViT-CNN experiments, we follow the same setup as Lv et al. (2024).

We conduct our experiments using both homogeneous and heterogeneous teacher-student pairs. In homogeneous pairs, the teacher and student models belong to the same architectural family, whereas in heterogeneous pairs, the teacher and student come from different architectural families. We evaluate our approach across a range of architecture families, including ResNet (He et al., 2015), Wide ResNet (WRN) (Zagoruyko & Komodakis, 2016), VGG (Simonyan & Zisserman, 2014), ShuffleNet-V1/V2 (Zhang et al., 2018; Ma et al., 2018), and MobileNetV2 (Sandler et al., 2018).

We evaluate the effectiveness of our approach against various feature based and logit based approaches including KD (Hinton et al., 2015), FitNet (Romero et al., 2015), RKD (Park et al., 2019), CRD (Tian et al., 2022), OFD (Heo et al., 2019), ReviewKD (Chen et al., 2021), TAKD (Mirzadeh et al., 2020), DKD (Zhao et al., 2022), MLLD (Jin et al., 2023) and CAT-KD (Guo et al., 2023).

# B    Datasets

CIFAR-100 (Krizhevsky & Hinton, 2009) is a benchmark dataset composed of 60,000 RGB images, each of size $32 \times 32$ pixels, evenly distributed across 100 classes. The dataset is split into 50,000 training images and 10,000 test images, with each class containing 600 images. ImageNet (Deng et al., 2009) is a large-scale

Table 7: Medical imaging datasets for clinical evaluation.

| Dataset | Modality | Task | # Classes | # Samples | Train/Val/Test |
|---------|----------|------|-----------|-----------|----------------|
| DermaMNIST | Dermatoscope | Multi-class | 7 | 10,015 | 7,007/1,003/2,005 |
| BreastMNIST | Ultrasound | Binary | 2 | 780 | 546/78/156 |
| OrganMNIST-S | CT (Sagittal) | Multi-class | 11 | 25,221 | 13,940/2,452/8,829 |

dataset widely used for image classification tasks. The dataset consists of over 1.2 million high-resolution training images and 50,000 validation images, categorized into 1,000 distinct classes.

### B.1 MedMNIST Subsets and Protocol

We select three representative medical imaging datasets from MedMNIST v2 (Yang et al., 2023): DermaM-NIST (dermatoscopic images, 7-class skin lesion diagnosis), BreastMNIST (breast ultrasound, binary tumor classification), and OrganMNIST-S (sagittal CT, 11-organ recognition). Table 7 summarizes the dataset characteristics.

We follow the standard MedMNIST training protocol with images resized to $224 \times 224$ pixels. ResNet-50 serves as the teacher model, while ShuffleNetV2 (SV2) and MobileNetV2 (MV2) are used as lightweight student models suitable for mobile deployment. We compare MCD against student-only training, traditional KD (Hinton et al., 2015), and DKD (Zhao et al., 2022). For MCD, we set $\alpha = 8$ and $\beta = 2$. All experiments are averaged over 3 runs.

## C   Teacher Training Statistics

The training accuracy for all teacher models across datasets are reported in Table 8. These statistics clarify the activation frequency of the ECKD branch: higher training accuracy means fewer teacher errors and thus less ECKD activation.

Table 8: Training accuracy of teacher models across all datasets. CIFAR-100 CNN teachers exhibit severe overfitting ($>99\%$), limiting ECKD activation. ImageNet, MedMNIST, and CIFAR-100 ViT teachers show meaningful error rates (3–15%) that activate the error-correction mechanism.

| Dataset | Teacher | Training Acc (%) | Error Rate (%) |
|---------|---------|------------------|----------------|
| CIFAR-100 (CNN) | ResNet56 | 97.91 | 2.09 |
| | ResNet110 | 99.54 | 0.46 |
| | ResNet32x4 | 99.97 | 0.03 |
| | ResNet50 | 99.96 | 0.04 |
| | WRN40-2 | 99.88 | 0.12 |
| | VGG13 | 99.96 | 0.04 |
| | ConvNeXt-Tiny | 93.08 | 6.92 |
| CIFAR-100 (ViT) | Swin-Tiny | 94.47 | 5.53 |
| ImageNet | ResNet34 | 84.69 | 15.31 |
| | ResNet50 | 87.46 | 12.54 |
| MedMNIST | ResNet50 (DermaMNIST) | 95.01 | 4.99 |
| | ResNet50 (OrganMNIST-S) | 97.23 | 2.77 |
| | ResNet50 (BreastMNIST) | 92.12 | 7.88 |

CIFAR-100 CNN teachers are severely overfitted, making gains primarily attributable to the AAKD branch. To enable some ECKD activation under such conditions, we retain teachers in training mode (Section 6.1). ImageNet teachers exhibit 12–15% error rates, providing substantial ECKD activation. ViT teacher on

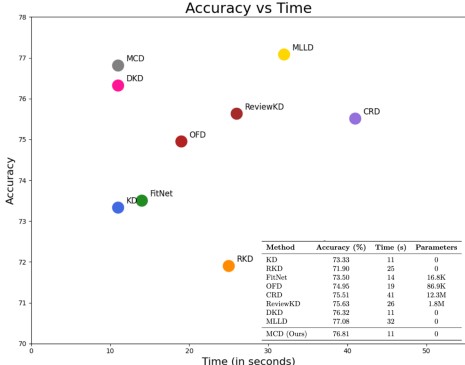

Figure 2: Performance of different KD techniques, with accuracy plotted on the y-axis and the corresponding computation time per batch on the x-axis on CIFAR-100 with teacher-student pair as ResNet32x4-ResNet8x4.

CIFAR-100 show moderate error rates (5%), and notably ConvNeXt-Tiny's higher error rate (6.92%) correlates with larger MCD improvement (+2.61% vs +1.48%). MedMNIST's data-scarce setting yields higher teacher error rates and correspondingly larger MCD gains.

# D    Additional Results

## D.1    Training Efficiency (No Param Overhead)

MCD retains the same time complexity as traditional Knowledge Distillation (KD), making it a highly efficient approach. Moreover, MCD achieves its performance improvements without introducing any additional parameters as shown in Figure 2, ensuring that the training process remains straightforward and resource-effective.

## D.2    Isolating AAKD vs. ECKD on ImageNet

Our earlier ablations (Table 5) examined the interplay between Accuracy-Aligned Knowledge Distillation (AAKD) and Error-Correction Knowledge Distillation (ECKD) by varying the sensitivity of the hyperparameters $\alpha$ and $\beta$. While those results highlighted the trade-off between the two components, they did not directly disentangle their individual contributions. To provide a clearer picture, we now conduct an additional study on ImageNet with a heterogeneous teacher–student pair (ResNet50 $\to$ MobileNetV1).

**Ablation Design.**    Table 9 shows the resulting loss functions. Crucially, both ablation configurations apply **identical loss to correct and incorrect samples**, effectively removing the gating mechanism and isolating each branch's contribution.

Table 9: Summary of ablation loss functions and calibrated top-1 accuracy on ImageNet with ResNet50→MV1. $\mathrm{Acc}_T$: accuracy when teacher is correct; $\mathrm{Acc}_{\neg T}$: accuracy when teacher is incorrect.

| Configuration | Correct Samples | Incorrect Samples | Acc. | $\mathrm{Acc}_T$ | $\mathrm{Acc}_{\neg T}$ |
|---|---|---|---|---|---|
| MCD (full) | $12 \cdot \mathcal{L}_{KD} + 1 \cdot \mathcal{L}_{CE}$ | $1 \cdot \mathcal{L}_{KD} + 2 \cdot \mathcal{L}_{CE}$ | 73.65 | 90.8 | 18.01 |
| ECKD-all | $1 \cdot \mathcal{L}_{KD} + 2 \cdot \mathcal{L}_{CE}$ | $1 \cdot \mathcal{L}_{KD} + 2 \cdot \mathcal{L}_{CE}$ | 68.09 | 83.8 | 17.9 |
| AAKD-all | $12 \cdot \mathcal{L}_{KD} + 1 \cdot \mathcal{L}_{CE}$ | $12 \cdot \mathcal{L}_{KD} + 1 \cdot \mathcal{L}_{CE}$ | 70.7 | 88.4 | 14.1 |

Table 9 reports the calibrated top-1 accuracy under three configurations: (i) the full Mentor-Critic Distillation (MCD), (ii) ECKD-all , and (iii) AAKD-all. The results yield several important observations. First, both AAKD and ECKD in isolation underperform the full MCD framework, indicating that neither component alone is sufficient to capture the full benefit of correctness-aware distillation. Second, AAKD-all improves

accuracy when the teacher is correct ($\text{Acc}_T$), but struggles significantly when the teacher is wrong ($\text{Acc}_{\neg T} = 14.1$), underscoring the risk of blindly following the teacher's knowledge. Conversely, ECKD-all provides stronger correction when the teacher is incorrect ($\text{Acc}_{\neg T} = 17.9$), but suffers from limited regularization when the teacher is correct, leading to reduced overall accuracy. Finally, MCD effectively combines these strengths, achieving the best overall performance (73.65%), the highest $\text{Acc}_T$ (90.8), and a competitive $\text{Acc}_{\neg T}$ (18.01).

These findings clearly demonstrate the complementary roles of AAKD and ECKD: AAKD leverages the teacher's reliability for regularization, while ECKD safeguards against negative transfer when the teacher errs. Their synergy within MCD enables robustness across both scenarios, validating the necessity of correctness-aware decoupling in knowledge distillation.

### D.3 Component Ablation on CIFAR-100

As described in Section 6.1, we retain teachers in training mode during distillation. For ResNet32x4, the teacher's prediction accuracy on the training set changes from 99.97% (eval mode) to 96.96% (train mode) due to dropout/BN stochasticity, so ECKD activates on approximately 3% of samples.

Table 10: Component ablation on CIFAR-100 (ResNet32x4 → ResNet8x4). All methods use identical training protocol: SGD with LR=0.05, momentum=0.9, weight decay=5e-4, following standard KD evaluation practice.

| Configuration | Loss Function | Accuracy |
|---|---|---|
| Vanilla KD ($\lambda = 0.9$) | $0.9 \cdot \mathcal{L}_{KD} + 0.1 \cdot \mathcal{L}_{CE}$ | 73.33% |
| AAKD-all (no gating) | $12 \cdot \mathcal{L}_{KD} + 1 \cdot \mathcal{L}_{CE}$ | 75.65% |
| MCD (with gating) | Eq. 6–7 | 76.81% |
| Gating contribution (MCD − AAKD-all) | | +1.16% |

Crucially, both AAKD-all and MCD use identical train-mode teachers, ensuring the +1.16% improvement is attributable solely to correctness-aware gating.

Although the KD/CE mixture can be renormalized to an equivalent form in principle, in practice changing KD/CE coefficients changes the gradient scale and interacts with the fixed optimizer/schedule (SGD momentum, weight decay, LR schedule). Following standard KD literature and benchmarks, we keep the optimizer configuration fixed across methods rather than renormalizing the objective and jointly retuning optimizer hyperparameters for all baselines.

