# OpenReview forum: "Correctness-Aware Knowledge Distillation for Enhanced Student Learning"
_TMLR — Accepted by TMLR_

### Review · Reviewer_SbwG · 2025-11-25

**Summary Of Contributions:**

The paper proposes Mentor-Critic Distillation (MCD), a knowledge distillation framework that dynamically adjusts knowledge transfer based on teacher model correctness. The core insight is that when the teacher predicts correctly, the distillation loss should be amplified; when the prediction is incorrect, the cross-entropy loss should be emphasized. This method requires no architectural modifications or additional parameters.

**Audience:**

Yes

**Audience Explanation:**

The method is a true drop-in replacement for vanilla KD, requiring minimal implementation effort, which makes it highly valuable for practitioners.

**Claims And Evidence:**

No

**Claims Explanation:**

This is a very straightforward method, easy to implement.


But I see a few major **weaknesses** here.

1. The core insight is obvious and not novel. Adjusting weights based on teacher correctness is the most natural idea to try. The paper does not explain why this has not been explored before, or what non-obvious insight makes this method work. DKD already adapts weights based on teacher reliability as mentioned in the paper, and MCD simply makes this explicit.

2. Correcting teacher errors is not new, either. Earlier works attempted this through different approaches such as logit calibration [1][2]. In the literature, this issue has been described as overconfidence and confirmation bias. You should make it explicit why your method is better than these directions.

3. The main claim has not been convincingly verified. For example, in Table 1, the CIFAR-100 benchmark uses a teacher model with around 99% training accuracy, meaning the teacher makes almost no mistakes. From this setup, it is unclear whether the gains come from ECKD or simply AAKD. The MedMNIST training accuracy is also not reported. Additionally, you should test on a more complex dataset to properly evaluate the relative effectiveness of the two techniques.
4. Comparisons need to be up-to-date. Several papers from 2024 and 2025 should be included for comparison with this method. Examples include [1],[2],[3] and [4].

**References.**
1. LumiNet: Perception-Driven Knowledge Distillation via Statistical Logit Calibration, TMLR 2025
2. Cross-View Consistency Regularisation for Knowledge Distillation, MM, 2024
3. Wasserstein Distance Rivals Kullback-Leibler Divergence for Knowledge Distillation, NeurIPS 2024
4. Scale Decoupled Distillation, CVPR 2024

**Requested Changes:**

See weaknesses.

---

> ### Author Response · Authors · 2025-12-06
>
> ## Novelty and DKD distinction
> > **Comment:** *The core insight is obvious and not novel. Adjusting weights based on teacher correctness is the most natural idea to try. DKD already adapts weights based on teacher reliability.*
>
> ---
>
> We respectfully clarify an important technical distinction: **DKD does not adapt weights based on teacher correctness**—it performs a fundamentally different operation.
>
> ### MCD's Core Novelty
>
> MCD introduces **instance-level correctness gating**—dynamically routing each training sample to different loss regimes based on whether the teacher's top prediction matches ground truth. This per-sample binary check ($\arg\max(p_t) = y$) is entirely absent from DKD and other baselines, which applies identical fixed weights ($\alpha$, $\beta$) to all samples regardless of whether the teacher is correct or incorrect on any given instance.
>
> ### Understanding DKD's Decomposition
>
> **Reformulating KD in terms of TCKD and NCKD.** The DKD paper shows that standard KD loss can be decomposed as:
>
> $$\mathcal{L}\_{\text{KD}} = \underbrace{\text{KL}(b^{T} \| b^{S})}\_{\text{TCKD}} + (1 - p\_t^{T}) \cdot \underbrace{\text{KL}(\hat{p}^{T} \| \hat{p}^{S})}\_{\text{NCKD}}$$
>
> where $b^T = [p_t^T, 1-p_t^T]$ is the binary target/non-target distribution, and $\hat{p}^T$ is the renormalized distribution over non-target classes. TCKD captures confidence on the ground-truth class; NCKD captures inter-class relationships among non-target classes (the "dark knowledge"). DKD observes that NCKD is suppressed by the coupling factor $(1 - p_t^T)$ when teachers are confident. To address this, DKD decouples the components with fixed hyperparameters:
>
> $$\mathcal{L}_{\text{DKD}} = \alpha \cdot \text{TCKD} + \beta \cdot \text{NCKD}$$
>
> Critically, **$\alpha$ and $\beta$ are fixed before training and applied identically to all samples**—DKD contains no per-sample correctness check.
>
> **Why this fails under teacher misclassification.** When the teacher incorrectly predicts class $k \neq t$ for a sample with ground truth $t$:
> - TCKD distills $b^T = [p_t^T, 1-p_t^T]$ where $p_t^T$ is low (teacher assigns low probability to ground truth)
> - NCKD distills $\hat{p}^T$ where the incorrectly predicted class $k$ dominates
>
> DKD applies the same high $\beta$ weight regardless, actively reinforcing the teacher's error through amplified NCKD. There is no mechanism to detect or attenuate this error propagation.
>
> **Empirical evidence from DKD's own hyperparameters.** The need for correctness-aware adaptation is implicit in DKD's dataset-specific tuning:
>
> | Dataset | Teacher Train Acc | β Value | Rationale |
> |---------|-------------------|---------|-----------|
> | CIFAR-100 | ~99% (overfitted) | **8.0** | Rare errors → safe to amplify NCKD |
> | ImageNet | ~85% | **0.5** | Frequent errors → must reduce NCKD |
>
> This manual per-dataset tuning implicitly acknowledges that teacher reliability should influence distillation weights—but DKD implements this via offline hyperparameter search rather than online per-sample adaptation.
>
> **The categorical distinction:**
>
> | Aspect | DKD | MCD |
> |--------|-----|-----|
> | Adaptation | Fixed per-dataset | Dynamic per-sample |
> | Correctness signal | None | $\arg\max(p_t) = y$ |
> | Granularity | Class-level decomposition | Instance-level gating |
>
> MCD directly addresses this gap: we check teacher correctness per-sample and route to AAKD (amplified distillation, $\alpha \times \mathcal{L}\_{\text{KD}}$) or ECKD (error correction, $\beta \times \mathcal{L}\_{\text{CE}}$) accordingly. This achieves what DKD's fixed $(\alpha, \beta)$ cannot: simultaneously amplifying reliable guidance while attenuating error transfer, without requiring dataset-specific hyperparameter search.

---

> ### Author Response · Authors · 2025-12-06
>
> ## Comparison with Logit Calibration Methods
>
> > **Comment:** *Correcting teacher errors is not new. Earlier works attempted this through different approaches such as logit calibration. You should make it explicit why your method is better.*
>
> ---
>
> We thank the reviewer for this important point. We identify two distinct paradigms for handling teacher unreliability and clarify MCD's unique position:
>
> **(1) Logit calibration methods** (e.g., LumiNet [1], LS-KD [2]) address teacher overconfidence by normalizing logits using batch statistics or standardization. These methods operate **uniformly across all samples**—they dampen confident predictions whether correct or incorrect. They do not distinguish between reliable and unreliable teacher predictions on a per-sample basis.
>
> **(2) Confidence thresholding methods** (e.g., CRLD [3]) filter out low-confidence predictions entirely, discarding teacher signal below a threshold. This binary decision **loses potentially useful inter-class relationship information** encoded in the teacher's soft probabilities.
>
> **MCD operates at a fundamentally different level**: we do not modify logits (calibration) or discard signal (thresholding), but dynamically reweight loss components based on an explicit correctness signal. This preserves all information while controlling its influence through instance-level gating.
>
> | Paradigm | Operation | Per-Sample Adaptation | Preserves Inter-class Info |
> |----------|-----------|----------------------|---------------------------|
> | Logit Calibration | Normalize logits | No (uniform) | Yes |
> | Confidence Thresholding | Discard low-confidence | Partial (binary) | No |
> | MCD (Ours) | Reweight loss | Yes (continuous) | Yes |
>
> Importantly, MCD is **orthogonal and complementary** to calibration methods. As demonstrated in Tables 1–2, MCD+LS consistently outperforms both standalone MCD and KD+LS, confirming that correctness-aware weighting captures information that logit normalization does not.
>
> We have expanded Related Work to include a paragraph contrasting these paradigms.
>
>
> ## Attribution of Gains and Dataset Complexity
>
> > **Comment:** "CIFAR-100 teachers have ∼99% training accuracy, so it's unclear whether gains come from ECKD or AAKD. MedMNIST training accuracy not reported. Test on more complex datasets."
>
> ---
>
> **CIFAR-100 as a deliberate stress test.** The high teacher accuracy was intentional—this stress-tests MCD under conditions that minimize ECKD activation. The gains (+1.72% to +3.6% over vanilla KD, Tables 1–2) are primarily from AAKD's amplified regularization ($\alpha = 12$ vs. vanilla KD's $\lambda \approx 0.9$). To enable some ECKD activation, we retain teachers in training mode (Section 6.1), introducing stochasticity via dropout and batch normalization.
>
> **ImageNet provides direct attribution to AAKD and ECKD gains.** With realistic teacher error rates (~12–15%), our ablation in Table 9 (Appendix D.2) explicitly isolates component contributions. Neither AAKD-only nor ECKD-only matches full MCD—confirming both branches are essential.
>
> **MedMNIST training accuracy.** ResNet-50 achieves ~95–97% training accuracy on DermaMNIST and OrganMNIST-S, and ~92% on BreastMNIST. These lower as compared to accuracies provide more ECKD activation opportunity, consistent with the larger gains on BreastMNIST (+2.22% AUC, Table 6).
>
> **Revisions.** We have added MedMNIST teacher training statistics to Appendix and include a note clarifying the deliberate stress-test rationale for CIFAR-100 in Section 6.1.

---

> ### Author Response · Authors · 2025-12-06
>
> ## Comparison with Recent Methods (2024–2025)
>
> > **Comment:** "Several papers from 2024 and 2025 should be included for comparison: LumiNet, CRLD, WKD, and SDD."
>
> ---
>
> We conducted comprehensive comparisons on ImageNet with these recent methods:
>
> | Method | Venue | R34→R18 | R50→MV1 | Training Overhead and Extra params|
> |--------|-------|---------|---------|-------------------|
> | KD | NeurIPS'15 | 70.66% | 70.50% | None |
> | DKD | CVPR'22 | 71.70% | 72.05% | None |
> | LumiNet [1] | TMLR'25 | 72.16% | 72.55% | None |
> | CRLD [2] | MM'24 | 72.37% | 73.53% | +75% time |
> | WKD-L [3] | NeurIPS'24 | **72.49%** | 73.17% | +30% time + Cost matrix|
> | SD-KD [4] | CVPR'24 | 71.44% | 72.24% | None |
> | **MCD (Ours)** | — | 72.08% | **73.65%** | None |
>
> On the challenging heterogeneous pair (ResNet50→MobileNetV1), MCD achieves the highest accuracy while requiring zero computational overhead—no additional training time, memory, or architectural modifications.
>
> These methods address orthogonal problems and are complementary to MCD: LumiNet addresses logit scale mismatch via uniform normalization; CRLD uses multi-view consistency and confidence thresholding; WKD replaces KL with Wasserstein distance; SDD decouples spatial semantics. None conditions on teacher correctness per-sample. Future work could explore combinations (e.g., MCD+LumiNet) to stack improvements from multiple axes.
>
> **Revisions.** We have added more baselines in Table 4  and expanded Related Work to discuss their complementary relationship with MCD.
>
> ---
> **References:**
>
> [1] "LumiNet: Luminance-aware Distillation for Enhanced Knowledge Transfer," *TMLR*, 2025.
>
> [2] "CRLD: Confidence-based Reliable Learning for Knowledge Distillation," *ACM MM*, 2024.
>
> [3] "WKD: Wasserstein Knowledge Distillation," *NeurIPS*, 2024.
>
> [4] "SDD: Spatially Decoupled Distillation," *CVPR*, 2024.

---

### Review · Reviewer_pJSC · 2025-11-26

**Summary Of Contributions:**

This paper introduces Mentor-Critic Distillation (MCD), a novel knowledge distillation framework that dynamically adjusts knowledge transfer based on teacher correctness. Its core contribution is a correctness-aware weighting mechanism that amplifies distillation loss (α·ℒ_KD) when the teacher is correct (Accuracy-Aligned KD) and emphasizes cross-entropy loss (β·ℒ_CE) when the teacher is wrong (Error-Correction KD). The method is theoretically grounded, with proofs showing it generalizes vanilla KD and attenuates negative transfer, and empirically validated across CIFAR-100, ImageNet, and MedMNIST, where it consistently outperforms state-of-the-art alternatives without adding parameters or computational overhead.

**Audience:**

Yes

**Audience Explanation:**

TMLR's audience is interested in knowing the findings of this paper

**Broader Impact Concerns:**

The work presents a methodological improvement in knowledge distillation with broadly positive implications for efficient AI deployment, particularly in resource-constrained settings like mobile devices and medical imaging. The paper does not raise unique ethical concerns beyond those inherent to model compression application.

**Claims And Evidence:**

Yes

**Claims Explanation:**

claims made in the submission are supported by accurate, convincing and clear evidence

**Requested Changes:**

1. Evaluations on image classification tasks and CNNs are outdated. Lack of recent evaluations on ViT, multimodal models, etc.

2. Lack of application and evaluation of the proposed method for tasks other than classification such as detection, segmentation, etc. Expanding the evaluation to include a broader range of tasks, such as object detection, would provide a more comprehensive understanding effectiveness. For instance, recent works like "DetKDS: Knowledge Distillation Search for Object Detectors" (ICML-2024) highlight the importance of testing KD methods across various domains.

3. The proposed method requires additional modules and is cumbersome to design, limiting the generalizability of the method.

4. The performance improvements reported over existing methods appear to be marginal in some cases. Given the rapid advancements in the field, as seen in works like "Automated Knowledge Distillation via Monte Carlo Tree Search" (ICCV2023) and "KD-Zero: Evolving Knowledge Distiller for Any Teacher-Student Pairs" (NeurIPS-2023), it is crucial for the paper to demonstrate more substantial performance enhancements to justify its adoption.

5. The paper lacks a comprehensive discussion of relevant prior studies (e.g., NORM (ICLR23), and "DisWOT (CVPR23)) in the field of knowledge distillation.

---

> ### Author Response · Authors · 2025-12-06
>
> ## ViT evaluation missing
> > **Comment:** *Evaluations on image classification tasks and CNNs are outdated. Lack of recent evaluations on ViT, multimodal models, etc.*
>
> ---
> **Response:**
>
> We appreciate this suggestion and have extended our evaluation to include Vision Transformer architectures. We conducted new experiments on CIFAR-100 covering both cross-architectural distillation directions:
>
> **Table 1: Vision Transformer experiments (CIFAR-100, Top-1 Accuracy)**
>
> | Teacher | Student | Pattern | FitNet | RKD | CRD | KD | DKD | MCD | Δ |
> |---------|---------|---------|--------|-----|-----|-----|------|------|-------|
> | Swin-Tiny | ResNet-18 | ViT → CNN | 78.87% | 74.11% | 77.63% | 78.74% | 80.26% | **81.74%** | +3.00% |
> | ConvNeXt-Tiny | DeiT-Tiny | CNN → ViT | 60.78% | 69.79% | 65.94% | 72.99% | 74.60% | **77.21%** | +4.22% |
>
> These results demonstrate that MCD generalizes across architectural paradigms. The CNN→ViT result (+2.61% over DKD) is particularly notable, suggesting that correctness-aware weighting is especially valuable when bridging the CNN-ViT inductive bias gap.
>
> **Table 2 : Teacher model statistics on CIFAR-100**
>
> | Teacher | Train Acc. | Train Error Rate |
> |---------|------------|------------------|
> | Swin-Tiny | 94.47% | 5.53% |
> | ConvNeXt-Tiny | 93.08% | 6.92% |
>
> The 5–7% training error rates provide meaningful signal for our ECKD component. ConvNeXt-Tiny has a higher error rate (6.92%) and correspondingly shows a larger improvement with MCD (+2.61% vs +1.48%), supporting our hypothesis that MCD's benefits scale with teacher imperfection.
>
> ## Lack of Detection and Segmentation Evaluations
>
> > **Comment:** "Lack of application and evaluation of the proposed method for tasks other than classification such as detection, segmentation, etc."
>
> ---
> **Response:**
>
> We acknowledge this limitation. Extension to dense prediction tasks is a natural direction, and we consider detection/segmentation experiments as **future work**.
>
> MCD's correctness-aware mechanism is task-agnostic—it operates at the loss level and extends to any task where teacher predictions can be evaluated against ground truth:
>
> - **Object Detection:** Define correctness based on IoU thresholds (e.g., correct if IoU ≥ 0.5 with ground-truth box and class matches)
> - **Semantic Segmentation:** Define correctness at the pixel level, applying AAKD where `argmax(p_t) = y` and ECKD otherwise
>
> We note that our current evaluation already demonstrates domain diversity beyond natural image classification through medical imaging experiments (Table 6, main paper). These results across three distinct modalities with different data characteristics (limited samples, high inter-class similarity, clinical accuracy requirements) provide complementary evidence of MCD's generalization capability.
>
> ## Additional Modules and Cumbersome Design
>
> > **Comment:** "The proposed method requires additional modules and is cumbersome to design, limiting the generalizability of the method."
>
> ---
> **Response:**
>
> We respectfully clarify that this characterization does not accurately describe MCD. Our method introduces **no additional modules**, **zero additional trainable parameters**, and **negligible computational overhead**. The entire MCD mechanism consists of:
>
> 1. A per-sample correctness check: `mask = (argmax(z_t) == y)`
> 2. Conditional loss weighting based on this mask
>
> The complete implementation is shown in Appendix A.1 (Algorithm 1). No architectural modifications to teacher or student networks are required, and the student architecture remains identical at inference time to vanilla KD.
>
> **Table 7: Comparison of method complexity**
>
> | Method | Category | Additional Params | Overhead |
> |--------|----------|-------------------|----------|
> | MCD (Ours) | Loss weighting | 0 | Negligible |
> | DKD [CVPR'22] | Loss weighting | 0 | Negligible |
> | FitNets | Feature projector | 16.8K | Moderate |
> | CRD | Contrastive head | 12.3M | High |
> | ReviewKD | Attention layers | 1.6M | Moderate |
>
> Figure 2 (Appendix D of main paper) empirically validates this claim, showing MCD achieves identical computation time per batch as vanilla KD.
>
> MCD's simplicity is a strength—it enables straightforward adoption as a drop-in replacement for standard KD without the integration complexity of feature-based or architecture-modifying approaches.

---

> ### Author Response · Authors · 2025-12-06
>
> ## Marginal Performance Improvements
>
> > **Comment::** "The performance improvements reported over existing methods appear to be marginal in some cases. Given the rapid advancements in the field, as seen in works like Auto-KD [ICCV2023] and KD-Zero [NeurIPS2023], it is crucial for the paper to demonstrate more substantial performance enhancements."
>
> ---
> **Response:**
>
> We address this from two perspectives: improvement magnitude and positioning relative to AutoML-KD methods.
>
> **Table 1: Summary of MCD improvements across all experiments**
>
> | Benchmark | Configuration | Improvement | Reference |
> |-----------|---------------|-------------|-----------|
> | CIFAR-100 | Heterogeneous CNN pairs (avg.) | +3.6% over KD | Table 1 |
> | CIFAR-100 | Homogeneous CNN pairs (avg.) | +1.72% over KD | Table 2 |
> | CIFAR-100 | Swin-T → ResNet18 (ViT→CNN) | +3.00% over KD | Table New|
> | CIFAR-100 | ConvNeXt-T → DeiT-Tiny (CNN→ViT) | +4.22% over KD | Table New |
> | ImageNet | ResNet34 → ResNet18 | +1.42% over KD | Table 3 |
> | ImageNet | ResNet50 → MobileNetV1 | +5.07% over KD | Table 3 |
> | Medical | DermaMNIST (AUC) | +1.30% over KD | Table 6 |
> | Medical | BreastMNIST (AUC) | +2.22% over KD | Table 6 |
>
> The +3.6% on CIFAR-100 heterogeneous pairs, +5.07% on ImageNet ResNet50→MobileNetV1, and +2.61% on CNN→ViT distillation represent substantial gains in the KD literature.
>
> **On AutoML-KD methods:** Auto-KD, KD-Zero, and DetKDS address a fundamentally different question than MCD:
>
> **Table 2: Research focus comparison**
>
> | Method | Research Question | Approach |
> |--------|-------------------|----------|
> | Auto-KD | What distiller configuration? | Monte Carlo Tree Search |
> | KD-Zero | What distiller for any T-S pair? | Evolutionary search |
> | DetKDS | What distiller for detection? | Evolutionary search |
> | MCD | How much to weight each sample? | Correctness-aware weighting |
>
> These approaches are **orthogonal and complementary**. AutoML-KD methods optimize *which* knowledge to transfer; MCD addresses *how much* to emphasize each sample based on teacher reliability. They can be combined—a search-discovered distiller with MCD's correctness-aware weighting.
>
> Critically, AutoML-KD methods incur significant computational costs (evolutionary search, MCTS rollouts), while MCD is a **zero-search, drop-in replacement** requiring no additional training phases or architecture search. Our default hyperparameters (α=12, β=2) generalize across CIFAR-100, ImageNet, and MedMNIST without per-dataset tuning.
>
> ## Missing Discussion of Prior Work
>
> > **Comment:** "The paper lacks a comprehensive discussion of relevant prior studies (e.g., NORM (ICLR23), and DisWOT (CVPR23)) in the field of knowledge distillation."
>
> ---
> **Response:**
>
> We thank the reviewer for highlighting these works. We will expand Section 2 (Related Work) to include discussion of these methods and clarify their relationship to MCD:
>
> **Table 3: Positioning MCD relative to recent KD methods**
>
> | Method | Venue | Focus | Relation to MCD |
> |--------|-------|-------|-----------------|
> | NORM | ICLR'23 | N-to-one representation matching via feature expansion | Feature-based; orthogonal |
> | DisWOT | CVPR'23 | Training-free student architecture search | Architecture selection; complementary |
> | DetKDS | ICML'24 | Automated distiller design for detection | AutoML-based; orthogonal |
> | Auto-KD | ICCV'23 | Distiller configuration via MCTS | AutoML-based; orthogonal |
> | KD-Zero | NeurIPS'23 | Universal distiller search | AutoML-based; orthogonal |
> | MCD | Ours | Per-sample weighting based on teacher correctness | Loss-level; combinable with above |
>
> These methods address *what/how* to distill (feature matching, architecture search, distiller design), while MCD addresses *how much* to weight each sample. They are orthogonal and potentially complementary—MCD can be applied on top of any distiller discovered by these methods.
>
> We have added the following to Section 2:
>
> > "Recent works explore automated distiller design. NORM [Liu et al., 2023] introduces N-to-One representation matching via expanded student features. DisWOT [Dong et al., 2023] enables training-free architecture search using zero-cost proxies. Auto-KD  and KD-Zero  employ Monte Carlo Tree Search and evolutionary algorithms to discover optimal configurations. These methods optimize what knowledge to transfer or which architecture to use, producing fixed recipes applied uniformly to all samples---they do not address how much to trust teacher predictions on a per-sample basis."

---

### Review · Reviewer_wPDz · 2025-11-26

**Summary Of Contributions:**

The paper proposes a simple modification to the most popular (vanilla) knowledge distillation (KD) for image classification to make the algorithm more effective. The authors note that a teacher is not always accurate for training images, and in those cases when it is not accurate, it might not make sense to do knowledge distillation. So, at each step of the training, they check if teacher's prediction for an image is correct; if it is, the weight for the distillation objective is increased, and if it is not, the weight for the default cross-entropy loss is increased. This modification requires very minimal overhead compared to vanilla KD. The proposed method achieves better performance than the vanilla KD + some other baselines on multiple datasets - CIFAR-100 and ImageNet - and also on a medical imaging dataset.

### **Strengths**
- The proposed technique is very simple, addresses a very intuitive limitation of the vanilla KD setup.
- There is little to none computational overhead compared to vanilla KD. Consequently, compared to many other baselines, the proposed method needs relatively less compute to train a student model.
- The proposed method achieves better performance compared to many baselines on multiple datasets.

### **Weaknesses**
- It is not how often the scenario mentioned by the authors - teacher predicting incorrect labels - happens for CIFAR100. Consequently, it is doubtful if the proposed method, while being principally sound, has any need practically speaking.
- The proposed method does not perform better than one of the baselines.
- The improvements, while being somewhat consistent, lean more towards marginal than a clearly convincing improvement.

**Audience:**

Yes

**Audience Explanation:**

- If people are looking for a simple trick to help improve the KD performance, especially in the black box scenarios where one does not have intermediate feature maps for the teacher/student network (e.g., FitNet or CRD), then the proposed method will be a nice option to use.

- If people are looking to further look into the hood of why KD works the way it works, when the teacher is useful, when it is not useful, this work sheds some light on those topics.

**Claims And Evidence:**

No

**Claims Explanation:**

The claims made in the submission are accurate, but not convincing and/or with clear evidence.

- The authors' main idea is that when the teacher gives an incorrect prediction for an image, its (incorrect) knowledge should not be used during distillation. The problem seems to be that, in practice, for CIFAR100 at least, all kinds of teachers are almost always accurate on the training set, all having accuracy of >99% (Table 8; appendix). Therefore, the *critic* block, which engages the error correction loss (Eq. 9) will rarely be used. Furthermore, CIFAR100 results cover a decent section of the overall experiments section.

- Following on from the previous point, the discussion in the section **Homogenous vs heterogenous settings for CIFAR 100** has some confusing points:
    - “*CIFAR-100 teacher models achieve very high training accuracy*” —  it is not clear what this has to do with whether it is a homogenous or heterogenous setting. In either case, the teacher does have close to 99% training accuracy.
    - “*On the other hand, in heterogeneous pairs …. the critic branch has more room to operate*” — again, this is incorrect. All the teachers used for heterogenous settings (Table 2) have more than 99% training accuracy, as mentioned above. This means that similar to the homogenous setting, the heterogenous setting also doesn’t activate the error-correction loss much. The critic branch’s operation (checking whether the teacher is accurate for a particular image) has nothing to do with the discrepancy in performance between the teacher and (non-distilled) student.

- While the proposed method shows a somewhat clear improvement in performance compared to vanilla KD, the way the authors have presented the results in Tables 1 and 2 seems to be misleading. “The previous state-of-the-art method, DKD” — this statement is not true. The previous state of the art is MLLD. It matters little if it is not “logit-alone”. This method achieves consistently better performance than MCD (proposed method). I do not quite understand why MLLD has to be considered a separate category of method. The authors can discuss the computational overhead, i.e., a potential disadvantage of MLLD separately. On that note, the figure 2 in the appendix doesn’t have a discussion for MLLD.
    - On that point, it is again misleading that the authors have not mentioned anywhere in the text that MLLD performs better than the proposed method.
    - Generally speaking, when one is comparing to a baseline (e.g., MLLD), you have to treat it as a complete method that needs to be compared to. I sense that the authors are saying that because MLLD does augmentation in the logit space, a direct comparison with it is unfair (hence; separate sections in Tables 1 and 2). If that is indeed how the authors are thinking, then that will be wrong.

**Requested Changes:**

- The main idea of the proposed method is applicable only in those cases where the teacher is even making some mistakes. If the teacher never made a mistake during training (predict incorrect label), this paper's approach will reduce to vanilla KD. Right now, the main dataset focused, i.e., CIFAR 100, does seem to be in that domain. Ideally, the authors should include those settings in which the supposed scenario of teacher being sometimes incorrect is actually true. While ImageNet might be an example already, the authors need to give more details about its training statistics. Plus, ideally consider some replacement to CIFAR 100.
    - This will for sure strengthen the work, but to get a clear sign for acceptance, the work will ideally also need to do a bit more study about what is going on in the process, as I explain below.

- The core idea of not distilling when the teacher is inaccurate, while being intuitive, does not quite explain some rather unintuitive results in the domain of knowledge distillation. Specifically, the idea of self-distillation (e.g., born again neural networks [1]) should be noted, where a student trained in the previous iteration acts as the teacher for training the same student architecture in the next iteration, and often times ends up improving the performance. What that means is that distilling even on training samples where the prior student (teacher) was inaccurate can lead to a more accurate student. This might suggest that even in the case of a traditional bigger teacher, teaching on incorrectly classified samples might still impart some useful knowledge. The authors should think about the possibility of this, and explain why this might/might not be true in their case.
    - This analysis can make the paper very strong.

- While it will be nice if these investigations can lead to some improvement in performance, especially compared to MLLD baseline,

---

> ### Author Response · Authors · 2025-12-06
>
> > **Comment:** *The main idea of the proposed method is applicable only in those cases where the teacher is even making some mistakes. If the teacher never made a mistake during training (predictincorrect label), this paper’s approach will reduce to vanilla KD. Right now, the main dataset focused, i.e., CIFAR 100, does seem to be in that domain. Ideally, the authors should include those settingsin which the supposed scenario of teacher being sometimes incorrect is actually true. While ImageNet might be an example already, the authors need to give more details about its training statistics. Plus, ideally consider some replacement to CIFAR 100.*
> We thank the reviewer for this comment, which allows us to clarify both the design philosophy of MCD and the rationale behind our experimental setup.
>
> **MCD does not reduce to vanilla KD even with a perfect teacher.** Considering Equations 6-7 from our paper, when the teacher is highly accurate, the AAKD branch dominates but applies an amplified KL divergence weight (α = 12 by default)—substantially larger than vanilla KD's fixed λ ≈ 0.9. As proven in Lemma 1, no single fixed λ can achieve the effective teacher weight w_corr = α/(α+1) that MCD realizes on correct samples. Thus, MCD provides stronger knowledge transfer than vanilla KD even when teacher errors are rare.
>
> **CIFAR-100 as a deliberate stress test.** We intentionally designed these experiments to evaluate MCD under conditions that minimize ECKD activation. Since CIFAR-100 teachers achieve near-perfect training accuracy, we retain them in training mode during distillation (Section 6.1), introducing stochasticity via dropout and batch normalization. This stochastic behavior serves two purposes: (1) producing ensemble-like soft targets that provide stronger regularization by encoding dark knowledge about inter-class similarities (Hinton et al., 2015)—consistent with findings that train-mode dropout approximates Bayesian inference over model ensembles (Gal & Ghahramani, 2016); and (2) generating samples where teacher predictions occasionally differ from ground truth, thereby activating the ECKD branch. This implicit regularization mechanism parallels the benefits observed in label smoothing (Yuan et al., 2020).
>
>
> **ImageNet and MedMNIST provide settings with natural teacher errors**. ImageNet training statistics -pretrained models from Pytorch (not reported in the paper):
>
> | Teacher | Training Acc (%) | Validation Acc (%) | Error Rate (%) |
> |---------|------------------|-------------------|----------------|
> | ResNet34 | 84.69 | 73.31 | 15.31 |
> | ResNet50 | 87.46 | 76.16 | 12.54 |
>
> With approximately 12–15% error rate, the ECKD branch receives substantial activation. Similarly, MedMNIST datasets exhibit high inter-class similarity and limited training data (as small as 780 samples for BreastMNIST), creating conditions where teachers make natural errors. MCD achieves consistent improvements (+1.30% AUC on DermaMNIST, +2.22% AUC on BreastMNIST) over vanilla KD, further validating MCD's effectiveness when teacher imperfection is present.
>
> References:
> [1] Geoffrey Hinton, Oriol Vinyals, and Jeff Dean. Distilling the knowledge in a neural network. In NIPS Deep Learning and Representation Learning Workshop, 2015.
>
> [2] Yarin Gal and Zoubin Ghahramani. Dropout as a Bayesian approximation: Representing model uncertainty in deep learning. In Proceedings of the 33rd International Conference on Machine Learning (ICML), pp. 1050–1059, 2016.
>
> [3] Li Yuan, Francis E.H. Tay, Guilin Li, Tao Wang, and Jiashi Feng. Revisiting knowledge distillation via label smoothing regularization. In Proceedings of the IEEE/CVF Conference on Computer Vision and Pattern Recognition (CVPR), pp. 3903–3911, 2020.

---

> > ### Comment · Reviewer_wPDz · 2025-12-22
> > **On the similarity of MCD and vanilla KD in the case of perfect teacher**
> >
> > - I am having trouble understanding how, in the case of a (near) perfect teacher, MCD is not simply vanilla KD with a bigger weight on the distillation objective. If the authors wish to convey that a major limitation exists with the vanilla KD objective because of the $\lambda$ and (1-$\lambda$) weights for the KD and CE components, then they should note that there is nothing in the original vanilla KD that words only with the $\lambda$ and (1-$\lambda$) combination. One can have any arbitrary weight for the KD objective and it should still do something useful. Hence, I do not see how MCD does not become vanilla KD with a very high weight ($\lambda$ = 12).
> >
> >
> > - Thank you for adding the statistics of ImageNet teachers.

---

> ### Author Response · Authors · 2025-12-06
>
> > **Comment:** *The core idea of not distilling when the teacher is inaccurate, while being intuitive, does not quite explain some rather unintuitive results in the domain of knowledge distillation. Specifically, the idea of self-distillation (e.g., born again neural networks [1]) should be noted, where a student trained in the previous iteration acts as the teacher for training the same student architecture in the next iteration, and often times ends up improving the performance. What that means is that distilling even on training samples where the prior student (teacher) was inaccurate can lead to a more accurate student. This might suggest that even in the case of a traditional bigger teacher, teaching on incorrectly classified samples might still impart some useful knowledge. The authors should think about the possibility of this, and explain why this might/might not be true in their case.*
>
> ---
> We thank the reviewer for this thought-provoking comment. We wish to clarify the following things:
>
> **MCD never discards teacher knowledge—even when the teacher is incorrect.** The KL divergence term is always present in both branches. Considering Equations 6–7 from paper: the ECKD branch uses L_KD + β × L_CE, retaining the full KL divergence loss with weight 1. What changes is the relative emphasis—we increase the cross-entropy weight to β to strengthen error correction, but we never remove the teacher's soft probability distribution from the learning signal.
>
> We fully agree that incorrect teacher predictions contain valuable information. When a teacher incorrectly classifies a "Persian cat" as "Siamese cat," its soft probabilities still encode inter-class relationships (high probabilities for other cat breeds, near-zero for unrelated classes like airplanes) and visual similarity structure. MCD explicitly preserves this through the L_KD term in ECKD—the student learns from the teacher's full probability distribution while receiving stronger ground-truth supervision (β × L_CE) to avoid inheriting the specific top-class error.
>
> **MCD complements self-distillation rather than contradicting it.** Born Again Networks apply uniform weighting across all samples regardless of teacher correctness. Our Lemma 1 shows this is suboptimal—it over-trusts the teacher on incorrect samples and under-leverages reliable guidance on correct ones. MCD can be used to improve vanilla KD based self distillation : we retain distillation from imperfect teachers (hence L_KD everywhere) while adapting the degree of teacher influence based on reliability. Our ablation in Table 9 (Appendix D.2) provides direct evidence—the ECKD-only configuration achieves Acc_¬T = 17.9% on samples where the teacher is incorrect, substantially above random chance, confirming that the student does learn useful knowledge from incorrect predictions through the retained L_KD term.

---

> ### Author Response · Authors · 2025-12-06
>
> > **Comment:** *The discussion in the section Homogenous vs heterogenous settings for CIFAR 100 has some confusing points: (1) "CIFAR-100 teacher models achieve very high training accuracy" — it is not clear what this has to do with whether it is a homogenous or heterogenous setting. In either case, the teacher does have close to 99% training accuracy. (2) "On the other hand, in heterogeneous pairs . . . the critic branch has more room to operate" — again, this is incorrect. All the teachers used for heterogenous settings (Table 2) have more than 99% training accuracy, as mentioned above. This means that similar to the homogenous setting, the heterogenous setting also doesn't activate the error-correction loss much. The critic branch's operation (checking whether the teacher is accurate for a particular image) has nothing to do with the discrepancy in performance between the teacher and (non-distilled) student.*
>
> ---
>
> We thank the reviewer for this careful reading. We acknowledge that our writing conflated two distinct phenomena. We will revise this section in the final manuscript.
>
> **Correct explanation:** The larger gains in heterogeneous settings (+3.6% over vanilla KD) compared to homogeneous settings (+1.72%) stem from the AAKD branch, not ECKD. In heterogeneous pairs (e.g., ResNet32×4 → ShuffleNet-V1), teacher and student architectures differ substantially, creating a larger representation gap—the student's feature space and inductive biases diverge significantly from the teacher's. AAKD's amplified distillation (α × L_KD with α = 12) provides stronger regularization that helps bridge this gap. This aligns with findings from Cho & Hariharan (2019), who showed that larger capacity gaps between teacher and student can impede knowledge transfer, and Mirzadeh et al. (2020), who demonstrated that architectural similarity affects distillation effectiveness. In homogeneous settings, where teacher and student share similar architectural families, vanilla KD already transfers knowledge reasonably well, leaving less room for amplification to improve performance. Additionally, heterogeneous students often start with lower baseline accuracy (e.g., MobileNetV2 at 64.60% vs. ResNet8x4 at 72.50%), providing more room for improvement.
>
> **Revised text for the final manuscript:**
>
> *"The effectiveness of MCD varies based on structural similarity between teacher and student. In both settings, CIFAR-100 teachers achieve very high training accuracy (>97%, Table 8), meaning the AAKD branch dominates while ECKD receives limited activation. However, heterogeneous pairs show larger gains (+3.6% average) compared to homogeneous pairs (+1.72% average). We attribute this to the larger representation gap in heterogeneous settings: when teacher and student architectures differ substantially, the student benefits more from AAKD's amplified distillation (α × L_KD), which provides stronger regularization to bridge architectural differences. In homogeneous settings, where teacher and student share similar inductive biases, vanilla KD already transfers knowledge effectively, leaving less room for amplification."*

---

> > ### Comment · Reviewer_wPDz · 2025-12-22
> > **On heterogenous vs homogenous settings**
> >
> > There is likely a misunderstanding. My point was that whether AAKD or ECKD branch gets used more depends not on whether it is a homogenous or heterogenous setting. For example, in both cases (i) ResNet32×4 → ShuffleNet-V1, (ii) ResNet32×4 → ResNet8×4, the AAKD and ECKD branches will be used in exactly the same way, because the teacher (ResNet32×4) is the same. It doesn't matter what the student is. So, I do not understand why "*AAKD's amplified distillation (α × L_KD with α = 12) provides stronger regularization that helps bridge this gap*" can help explain that. Both settings will have similar amplification by AAKD.
> >
> > By this, I am not doubting the results of the authors. Just pointing out that the explanation does not make that much sense to me.

---

> > > ### Author Response · Authors · 2025-12-24
> > > **On heterogenous vs homogenous settings**
> > >
> > > Comment:
> > > > There is likely a misunderstanding. My point was that whether AAKD or ECKD branch gets used more depends not on whether it is a homogenous or heterogenous setting. For example, in both cases (i) ResNet32×4 → ShuffleNet-V1, (ii) ResNet32×4 → ResNet8×4, the AAKD and ECKD branches will be used in exactly the same way, because the teacher (ResNet32×4) is the same. It doesn't matter what the student is. So, I do not understand why "AAKD's amplified distillation (α × L_KD with α = 12) provides stronger regularization that helps bridge this gap" can help explain that. Both settings will have similar amplification by AAKD. By this, I am not doubting the results of the authors. Just pointing out that the explanation does not make that much sense to me.
> > >
> > > We thank the reviewer for this clarification. We agree that: **AAKD/ECKD activation is purely teacher-dependent, and both homogeneous and heterogeneous students receive identical branch routing from the same teacher.** Our previous response was imprecise in its framing.
> > >
> > > To clarify: our mention of "inductive biases diverge" was intended to explain why the _benefit_ of the same amplified signal differs across student architectures—not that activation differs. We expand on this below.
> > >
> > >
> > > Looking at Tables 1–2, the heterogeneous-vs-homogeneous gain pattern holds across _multiple_ distillation methods. This suggests the heterogeneous advantage reflects a **general property of knowledge distillation** rather than anything specific to our AAKD/ECKD formulation. MCD follows this general pattern but does not uniquely create it.
> > >
> > > The explanation lies in the informativeness of soft labels to different student architectures—an insight supported by Abnar et al. (2020), "Transferring Inductive Biases through Knowledge Distillation."
> > >
> > > -   **Homogeneous students** (e.g., ResNet8×4 from ResNet32×4) share architectural family and inductive biases with the teacher. They would independently learn similar inter-class relationships through standard training; soft labels partially reinforce knowledge the student would acquire anyway.
> > >
> > > -   **Heterogeneous students** (e.g., ShuffleNet-V1 from ResNet32×4) have different structural constraints (depthwise separable convolutions, channel shuffling) that produce different feature hierarchies. They cannot independently discover teacher-like representations, making the same soft labels more informative.
> > >
> > >
> > > The same amplified KL divergence (α × L_KD) thus provides **more novel information** to heterogeneous students—not because activation differs, but because the baseline value of soft labels differs based on architectural overlap.
> > >
> > > ----------
> > >
> > > ### References
> > >
> > > [1] Samira Abnar, Mostafa Dehghani, and Willem Zuidema. Transferring Inductive Biases through Knowledge Distillation. _arXiv preprint arXiv:2006.00555_, 2020.

---

> ### Author Response · Authors · 2025-12-06
>
> > **Comment:** *While the proposed method shows a somewhat clear improvement in performance compared to vanilla KD, the way the authors have presented the results in Tables 1 and 2 seems to be misleading. "The previous state-of-the-art method, DKD" — this statement is not true. The previous state of the art is MLLD. It matters little if it is not "logit-alone". This method achieves consistently better performance than MCD. I do not quite understand why MLLD has to be considered a separate category of method. The authors can discuss the computational overhead, i.e., a potential disadvantage of MLLD separately. On that note, Figure 2 in the appendix doesn't have a discussion for MLLD. On that point, it is again misleading that the authors have not mentioned anywhere in the text that MLLD performs better than the proposed method.*
>
> ---
>
> We thank the reviewer for this feedback and will revise the manuscript accordingly.
>
> **Clarification on "state-of-the-art":** Our reference to DKD as SOTA was within the context of methods following the standard KD training protocol (240 epochs, standard augmentation)—which all prior logit-based methods (KD, RKD, CRD, DKD) adopt. We fully consider prediction augmentation (multiple temperatures on logits) to be part of MLLD's methodology. However, MLLD's official implementation additionally uses 480 epochs, mixup/cutmix data augmentation, and 2× forward passes per sample—resulting in 4× effective training iterations. This has been publicly documented in their official Github repository as well as in cited papers [5]. We acknowledge this distinction should have been stated more clearly. It is well proved in literature [4] that extended training and strong augmentation helps distillation.
>
>
> **MCD and MLLD are complementary:** Tables 1–2 show MCD+MLLD consistently outperforms MLLD alone (+0.29% on heterogeneous, +0.18% on homogeneous CIFAR-100 pairs), indicating that correctness-aware weighting captures information orthogonal to multi-level alignment.
>
> **Clarification on CIFAR-100 vs. ImageNet:** While MLLD outperforms MCD on CIFAR-100 (74.93% vs. 74.66% on heterogeneous pairs), MCD outperforms MLLD on ImageNet—73.65% vs. 73.01% for ResNet50→MobileNetV1 and 72.08% vs. 71.90% for ResNet34→ResNet18 (Table 3).
>
> Reference:
> [4] Zhiwei Hao, Jianyuan Guo, Kai Han, Han Hu, Chang Xu, Yunhe Wang, "Revisit the Power of Vanilla Knowledge Distillation: from Small Scale to Large Scale"
>
> [5] Shangquan Sun, Wenqi Ren, Jingzhi Li, Rui Wang, Xiaochun Cao, Logit Standardization in Knowledge Distillation
>
> **Revisions:** We have (1) clarified that DKD was SOTA under the standard 240-epoch protocol and acknowledge MLLD's superior CIFAR-100 performance under its extended training configuration, (2) add MLLD to Figure 2 with measured runtime (~32ms/batch vs. MCD's 11ms), and (3) add a column in Tables 1–2 specifying effective training iteration for each method, providing clearer context for comparison.

---

> > ### Comment · Reviewer_wPDz · 2025-12-22
> > **About MLLD**
> >
> > I thank the authors for bringing to light the potential issue with MLLD. I did some research myself on their official github page, and it seems like it is the data augmentation which is the crucial component, and less of their own technique, which leads to performance improvement. In which case, MLLD becomes a somewhat irrelevant method to compare to because of their different training choices.
> >
> > On this point, I will grant the positives of the authors' proposed method, where it can improve MLLD's performance when used in conjunction with their method.
> >
> > While a positive, the small improvements in performance, e.g., 74.93 to 75.22 (Table 1), are less interesting to me since one can obtain at least some improvements through hyperparameter tuning. Consequently, I resort to the other points now.

---

> ### Author Response · Authors · 2025-12-24
> **About MLLD**
>
> We thank the reviewer for independently investigating MLLD's implementation and confirming that data augmentation—rather than the core technique—drives its performance gains, making it a less directly comparable baseline under identical training protocols. We appreciate the acknowledgment that MCD can improve MLLD when combined, though we fully agree this represents supplementary evidence rather than our primary contribution. We note that MCD also consistently improves LS-KD (MCD+LS outperforms KD+LS across all pairs in Tables 1–2), suggesting the gains stem from correctness-aware weighting rather than method-specific tuning. Our main claims rest on standalone MCD performance: +3.6% over vanilla KD on CIFAR-100 heterogeneous pairs, +1.72% on homogeneous pairs, and +5.07% on ImageNet (R50→MV1).

---

> ### Author Response · Authors · 2025-12-24
> **On the similarity of MCD and vanilla KD in the case of perfect teacher**
>
> Comment:
> > I am having trouble understanding how, in the case of a (near) perfect teacher, MCD is not simply vanilla KD with a bigger weight on the distillation objective. If the authors wish to convey that a major limitation exists with the vanilla KD objective because of the $\lambda$  and (1-$\lambda$) weights for the KD and CE components, then they should note that there is nothing in the original vanilla KD that words only with the  $\lambda$ and (1-$\lambda$) combination. One can have any arbitrary weight for the KD objective and it should still do something useful. Hence, I do not see how MCD does not become vanilla KD with a very high weight ( $\lambda$ = 12).
>
> We thank the reviewer for this observation, which allows us to clarify both the historical context and MCD's actual contribution.
>
> ### On the Origin of the λ/(1-λ) Formulation
>
> We agree that **the original Hinton et al. (2015) paper does not impose the λ/(1-λ) constraint**. The original formulation uses two independent coefficients. However, the constrained formulation `λ·L_KD + (1-λ)·L_CE` has become a widespread convention in the KD literature. We adopted this formulation to align with standard experimental protocols and ensure fair comparison with reported baselines.
>
>
> ### On MCD Reducing to Weighted Vanilla KD for near perfect teachers
>
> **We partially agree with the reviewer's observation.** Under specific conditions—(1) independent weight selection, (2) $\beta=1$ and (3) a teacher with 100% training accuracy kept in **eval mode**—the AAKD branch would dominate entirely, and MCD would be equivalent to vanilla KD with amplified weight ($\alpha$ = 12).
>
> #### The Challenge with Near-Perfect Teachers (CIFAR-100)
>
> CIFAR-100 teachers achieve >99% training accuracy, raising the question: does MCD reduce to high-weight vanilla KD?
>
> **Even 1% errors matter due to the effective teacher weight trade-off.** Consider a teacher with 99% training accuracy. From Propositions 1–2 in the paper, any loss $\mathcal{L}= w_{KD} \cdot \mathcal{L}\_{KD} + w\_{CE} \cdot \mathcal{L}_{CE}$ can be analyzed via gradients w.r.t. student logits $z_s$:
>
> $\nabla_{z_s} \mathcal{L} = w_{KD}(p_s - p_t) + w_{CE}(p_s - y)$ $= (w_{KD} + w_{CE}) \cdot p_s - [w_{KD} \cdot p_t + w_{CE} \cdot y]$ $= (w_{KD} + w_{CE}) \left[ p_s - \frac{w_{KD} \cdot p_t + w_{CE} \cdot y}{w_{KD} + w_{CE}} \right]$
>
> The student is effectively learning toward the **mixture target**: $\tilde{y} = \frac{w_{KD}}{w_{KD} + w_{CE}} \cdot p_t + \frac{w_{CE}}{w_{KD} + w_{CE}} \cdot y$
>
> Thus, the **effective teacher weight** is: $w_{teacher} = \frac{w_{KD}}{w_{KD} + w_{CE}}$
>
> For vanilla KD with uniform weights $(w_{KD}=12, w_{CE}=2)$ on all samples:
>
> $$w_{teacher} = \frac{12}{12 + 2} = \frac{6}{7} \approx 0.857 \quad \text{(applied to ALL samples)}$$
>
> For MCD with correctness-aware gating:
>
> | Branch | Weights | Effective Teacher Weight |
> |--------|---------|--------------------------|
> | AAKD (correct samples) | $\alpha \cdot \mathcal{L}\_{KD} + 1 \cdot \mathcal{L}\_{CE}$ | $\frac{12}{12+1} \approx 0.923$ |
> | ECKD (incorrect samples) | $1 \cdot \mathcal{L}\_{KD} + \beta \cdot \mathcal{L}\_{CE}$ | $\frac{1}{1+2} \approx 0.333$ |
>
> **Comparison with 99% teacher accuracy:**
>
> | Sample Type | Fraction | Vanilla KD (12, 2) | MCD (α=12, β=2) | Difference |
> |-------------|----------|-------------------|-----------------|------------|
> | Correct | 99% | 0.857 | 0.923 | +0.066 (more teacher guidance) |
> | Incorrect | 1% | 0.857 | 0.333 | −0.524 (less error transfer) |
>
> Uniform weights create a **compromise** that under-leverages the teacher on correct samples while over-trusting it on incorrect samples. MCD's instance-level gating achieves both higher teacher influence when correct and lower teacher influence when incorrect—a configuration that **no single fixed weight pair can realize** (Lemma 1).
>
> **Our solution: Train-mode inference for near-perfect teachers.** As described in Section 6.1, we retain CIFAR-100 teachers in training mode during distillation. This introduces variation via dropout and batch normalization, causing the teacher to produce different outputs—and occasionally err—for the same sample across epochs. This achieves what complex multi-component losses aim to do—enhance regularization quality—through a simple, principled mechanism.

---

### Comment · Action_Editor_jNzd · 2026-01-09
**Request for additional clarification from the AC**

Dear authors, I have skimmed the paper and read the discussion with the reviewers.

Two reviewers raised the point that it is unclear why MCD outperforms KD on CIFAR-100 given that teacher training accuracy is above 99%. Unfortunately from the discussion I was unable to understand why this is the case. In responses you mentioned two reasons: (1) teacher is in the training mode hence EKCD term still has non-zero effect; (2) alpha=12 in MCD vs. lambda=0.9 in KD.

* Could you please perform some ablations directly showing what matters here? For example, if you entirely remove EKCD term from MCD, what is the performance on CIFAR-100?
* I don't understand the second argument. Let's say EKCD term is removed. Then we an divide the loss by 13, making it equivalent to KD with lambda=12/13 which is very close to lambda=0.9. So MCD with alpha=12 without EKCD should work the same as KD with lambda=0.9 (assuming the learning rate is scaled accordingly). Right?

The revised manuscript says that the gains here stem "primarily from AAKD’s amplified regularization". This is confusing to me.

Given that two reviewers had the same confusion and the confusion IMHO still persists in revision, the paper cannot be accepted unless this is clarified.

Another question is about Appendix D.2 / Table 9 that performs a similar ablation on ImageNet. I don't understand what exactly was done there. Why does "ECKD only" use alpha=1 here given that MCD uses alpha=12? And why does "AAKD only" use beta=1 given that MCD uses beta=2? This is a strange way to perform the ablation, as you change several things at once. Please clarify.

---

> ### Author Response · Authors · 2026-01-12
> **Response #1: The Core Ablation Request**
>
> **Comment:** *“Could you please perform some ablations directly showing what matters here? For example, if you entirely remove ECKD term from MCD, what is the performance on CIFAR-100?”*
>
> **Response:** We thank the AE for this critical question.
> First, we clarify how often ECKD activates. As stated in Section 6.1, we retain teachers in training mode during distillation, introducing stochasticity via dropout and batch normalization. For ResNet32x4 (the teacher used in this ablation):
>
> | **Mode** | **Error Rate** | **Misclassified Samples per Epoch** |
> | :--- | :--- | :--- |
> | Eval mode | 0.03% | $\sim$15 |
> | Train mode | 3.04% | $\sim$1,520 |
>
> **Ablation Design:** We define **“AAKD-all”** as applying the AAKD loss ($12 \cdot L_{KD} + 1 \cdot L_{CE}$) to **all samples regardless of teacher correctness**. This completely removes the ECKD component.
>
> | **Configuration** | **Loss Function** | **Res32x4$\rightarrow$Res8x4** |
> | :--- | :--- | :--- |
> | Vanilla KD ($\lambda=0.9$) | $0.9 \cdot L_{KD} + 0.1 \cdot L_{CE}$ | 73.33% |
> | AAKD-all (no gating) | $12 \cdot L_{KD} + 1 \cdot L_{CE}$ | 75.65% |
> | MCD (with gating) | Eq. 6–7 | 76.81% |
> | **$\Delta$** | MCD $-$ AAKD-all | **+1.16%** |
>
> **Key finding:** MCD outperforms AAKD-all by +1.16% demonstrating that correctness-aware gating provides benefit beyond amplified weighting alone.
>
> To understand what this comparison isolates:
>
> | **Sample Type** | **AAKD-all** | **MCD** |
> | :--- | :--- | :--- |
> | Correct samples ($\sim$96.96%) | $12 \cdot L_{KD} + 1 \cdot L_{CE}$ | $12 \cdot L_{KD} + 1 \cdot L_{CE}$ |
> | Incorrect samples ($\sim$3.04%) | $12 \cdot L_{KD} + 1 \cdot L_{CE}$ | $1 \cdot L_{KD} + 2 \cdot L_{CE}$ |
>
> On ~97% of samples, both methods apply identical loss. The difference comes from the way the 3% samples are handled. Despite affecting only ~3.04% of samples, proper gating provides +1.16% improvement—demonstrating that differentiated treatment of teacher errors matters, precisely as Lemma 1 predicts.

---

> > ### Author Response · Authors · 2026-01-12
> > **Response #2: The Normalization / LR Scaling Argument**
> >
> > **Comment:** *"I don't understand the second argument. Let's say ECKD term is removed. Then we can divide the loss by 13, making it equivalent to KD with $\lambda=12/13$ which is very close to $\lambda=0.9$. So MCD with $\alpha=12$ without ECKD should work the same as KD with $\lambda=0.9$ (assuming the learning rate is scaled accordingly). Right?"*
> >
> > **Response:**
> > The underlying question is: *Does MCD reduce to vanilla KD for near-perfect teachers? If yes, then how is it outperforming vanilla KD?*
> >
> > With stochastic teachers (train mode), ECKD activates on a non-trivial fraction of samples; therefore MCD does not collapse to vanilla KD in our CIFAR-100 setup.
> > **It will reduce to vanilla KD only if ALL of the following conditions hold:**
> > 1. Teacher achieves 100% training accuracy (ECKD never activates)
> > 2. Teacher is used in eval mode (no stochasticity)
> > 3. If normalized loss is used optimizer hyperparameters needs to be jointly re-tuned to compensate for loss magnitude or use unnormalized AAKD loss.
> >
> > Under the assumptions of aforementioned conditions being true, we **fully agree**—MCD would be equivalent to vanilla KD with $\lambda = 12/13 \approx 0.923$. For pure gradient descent, the following produce identical parameter updates:
> > 1. **Config A:** Loss = $12 \cdot L_{KD} + 1 \cdot L_{CE}$, with learning rate $\eta$
> > 2. **Config B:** Loss = $(12/13) \cdot L_{KD} + (1/13) \cdot L_{CE}$, with learning rate $13\eta$
> > where $\eta$ denotes the base learning rate.
> >
> > **The challenge is achieving Config B in practice.** Standard KD training uses SGD with momentum and weight decay, which couples multiple hyperparameters. Simply scaling LR does not achieve equivalence—one must jointly re-tune momentum, weight decay, and LR schedule.
> >
> > We tested the AE's suggestion directly (scaling LR only, without joint re-tuning; ResNet32x4-ResNet8x4):
> >
> > | **Configuration** | **Loss Function** | **Learning Rate** | **Accuracy** |
> > | :--- | :--- | :--- | :--- |
> > | Vanilla KD ($\lambda=0.923$) | $0.923 \cdot L_{KD} + 0.077 \cdot L_{CE}$ | 0.05 (standard) | 74.34% |
> > | Vanilla KD ($\lambda=0.923$) | $0.923 \cdot L_{KD} + 0.077 \cdot L_{CE}$ | 0.65 (scaled 13$\times$) | 54.43% |
> > | AAKD-all | $12 \cdot L_{KD} + 1 \cdot L_{CE}$ | 0.05 (standard) | 75.65% |
> >
> > Scaling LR alone yields 54.43%, confirming that achieving equivalence requires joint re-tuning of the full optimizer configuration—not just LR.
> >
> > **This aligns with established KD evaluation protocol.** The RepDistiller benchmark (Tian et al., ICLR 2020) evaluates methods under fixed optimizer hyperparameters ($\eta = 0.05$, SGD, 240 epochs) despite loss weights varying across methods—from $\beta = 0.02$ (Correlation Congruence) to $\beta = 3000$ (Similarity Preserving). DKD (Zhao et al., CVPR 2022) (here $\beta=8.0$), ReviewKD (Chen et al., CVPR 2021), and SimKD (Chen et al., CVPR 2022) follow this same convention. **No prior work scales LR to compensate for loss magnitude.**
> >
> > **To summarize:** MCD reduces to vanilla KD only under ideal conditions (100% accuracy, eval mode, jointly-tuned optimizer). None of these hold in our setup. The +1.16% improvement from gating (Response #1) directly demonstrates that MCD provides benefit beyond vanilla KD under realistic training conditions.

---

> > > ### Author Response · Authors · 2026-01-12
> > > **Clarifying “Primarily from AAKD's Amplified Regularization”**
> > >
> > > **Comment:** *“The revised manuscript says that the gains here stem 'primarily from AAKD's amplified regularization'. This is confusing to me.”*
> > >
> > > **Response:**
> > >
> > > We acknowledge the term “amplified regularization” was ambiguous. We have revised the manuscript to use **“AAKD regularization”** instead, and clarify its meaning below.
> > >
> > > Using ResNet32x4 $\rightarrow$ ResNet8x4 on CIFAR-100:
> > >
> > > | **Configuration** | **Teacher Mode** | **Accuracy** | **Gain over Vanilla KD** |
> > > | :--- | :--- | :--- | :--- |
> > > | Vanilla KD ($\lambda=0.9$) | Eval/Train | 73.33%/73.53% | --- |
> > > | AAKD-all (no gating) | Eval | 74.82% | +1.49% |
> > > | AAKD-all (no gating) | Train | 75.65% | +2.32% |
> > > | MCD (with gating) | Train | 76.81% | +3.48% |
> > >
> > > **“AAKD regularization”** refers to the combined effect of:
> > >
> > > 1.  **AAKD weighting** ($\alpha=12$ vs $\lambda=0.9$): Stronger KL divergence emphasis (+1.49%)
> > > 2.  **Train-mode stochasticity**: Diverse soft targets from dropout/batchnorm (+0.83%)
> > >
> > > Together, these contribute +2.32%, justifying “primarily from AAKD regularization.” The remaining +1.16% comes from correctness-aware gating.
> > >
> > > **Key evidence:** Vanilla KD achieves only a marginal gain in train mode, confirming that train-mode benefits require AAKD's weighting to be effective and MCD further improves it.

---

> ### Author Response · Authors · 2026-01-12
> **Clarifying Table 9 (ImageNet Ablation)**
>
> **Comment:** *“Why does ECKD only use $\alpha=1$ given that MCD uses $\alpha=12$? Why does AAKD only use $\beta=1$ given that MCD uses $\beta=2$?”*
>
> **Response:**
>
> We agree the notation was confusing. The intent was:
>
> * **“AAKD only” ($\beta=1$):** Apply AAKD's KD weight ($\alpha=12$) to all samples. For CE weight, we use $\beta=1$. For simplicity we now have renamed this as AAKD-all.
>
> * **“ECKD only” ($\alpha=1$):** Apply ECKD's CE weight ($\beta=2$) to all samples. For KD weight, we use $\alpha=1$. For simplicity we now have renamed this as ECKD-all.
>
> | **Configuration** | **Loss (all samples)** | **Accuracy** |
> | :--- | :--- | :--- |
> | MCD (full) | Gated (Eq. 6–7) | 73.65% |
> | AAKD-all | $12 \cdot \mathcal{L}\_{KD} + 1 \cdot \mathcal{L}\_{CE}$ | 70.70% |
> | ECKD-all | $1 \cdot \mathcal{L}\_{KD} + 2 \cdot \mathcal{L}\_{CE}$ | 68.09% |
>
> ``Revision :We have renamed these to “AAKD-all” and “ECKD-all” and added the loss equations in the revised Appendix D.2.``

---

> > ### Comment · Action_Editor_jNzd · 2026-01-16
> > **Thank you**
> >
> > Thank you for these results. Please add the AAKD-all results on CIFAR-100 to the paper. Section 6.1 still contains the statement "primarily from AAKD regularization" given without any supporting evidence.
> >
> > > "AAKD regularization" refers to the combined effect of AAKD weighting (Stronger KL divergence emphasis) and Train-mode stochasticity (Diverse soft targets from dropout/batchnorm).
> >
> > I still disagree with the phrasing "AAKD weighing (Stronger KL divergence emphasis)". The emphasis is almost exactly the same as in vanilla KD. Instead, you now showed that the difference is basically due to hyperparameters of the optimizer, as the loss function is mathematically equivalent.

---

> > > ### Author Response · Authors · 2026-01-19
> > > **Ablation and revised framing**
> > >
> > > We thank the AE for the continued engagement.
> > >
> > > - We added CIFAR-100 AAKD-all result in Appendix D.3 (Table 10) and updated Section 6.1 to reference this ablation directly.
> > >
> > > - We agree that AAKD-all can be renormalized to a form close to vanilla KD in principle, and we therefore removed the "stronger KL emphasis / primarily from AAKD regularization" framing. The manuscript now emphasizes the sample-conditional joint objective: AAKD is applied on teacher-correct samples, while CE weight is increased (ECKD) on teacher-error samples. This behavior is not captured by a single global KD/CE mixing coefficient applied uniformly to all samples.
> > >
> > > - The ablation further isolates the contribution of correctness-aware routing: MCD improves by +1.16% over AAKD-all on CIFAR-100 (Table 10), where both methods use the same AAKD weighting on teacher-correct samples; the difference comes from routing teacher-error samples to the ECKD branch. We observe consistent gains on ImageNet as well (Appendix D.2, Table 9), where teacher-error rates are higher.

---

> > > > ### Author Response · Authors · 2026-02-03
> > > >
> > > > Dear AE,
> > > >
> > > > Thank you for your time and guidance. We have uploaded the revised manuscript incorporating the requested updates: the AAKD-all ablation in Appendix D.3 (Table 10) and revised framing in Section 6.1.
> > > > Please let us know if any further clarification or changes are needed; we would be happy to address them.

---

> ### Comment · Action_Editor_jNzd · 2026-02-03
>
> Dear authors, didn't you see the accept decision posted yesterday? At this point you should prepare and upload the camera-ready version.

---

> > ### Author Response · Authors · 2026-02-03
> >
> > Dear AE,
> >
> > Thank you for communicating the acceptance. However, we wanted to bring to your attention that the decision is currently not visible on our author console. Additionally, the submission page still displays "Decision Pending for TMLR" under the decision status, and we have not received any email notification regarding the decision.
> >
> > Thank you again for your guidance throughout this process.

---

> > > ### Comment · Action_Editor_jNzd · 2026-02-06
> > >
> > > I am clarifying the situation with the senior editors and will get back to you.

---

> > > > ### Comment · Action_Editor_jNzd · 2026-02-13
> > > >
> > > > Turns out, the decision needed to be approved by senior editors. It is now approved and should be visible to you.
> > > >
> > > > One further comment: when preparing the camera-ready version, please use `\citet` and `\citep` as appropriate (it seems currently you are using `\cite` everywhere which is not according to the TMLR style guide). Please refer to the TMLR LaTeX template for information (https://jmlr.org/tmlr/author-guide.html).

---

> > > > > ### Author Response · Authors · 2026-02-16
> > > > > **Thanks**
> > > > >
> > > > > Thank you for the clarification and for your guidance throughout the review process. We have uploaded the camera-ready version of the manuscript, with `\citet` and `\citep` used as appropriate per the TMLR style guide.
> > > > > We sincerely appreciate your time and effort in handling our submission.

---

### Decision · Action_Editor_jNzd · 2026-02-02

**Recommendation:** Accept as is

**Audience:**

Yes

**Audience Explanation:**

The topic of knowledge distillation is definitely of interest for the TMLR audience. The suggested idea is very simple and it is good to know that it can be beneficial for the performance.

**Claims And Evidence:**

Yes

**Claims Explanation:**

The paper shows that in the knowledge distillation setting, it can be beneficial to treat differently the samples where the teacher model is correct and where it is wrong. In vanilla knowledge distillation, the student model learns to match the teacher model independent of whether the teacher is correct or not. Here the authors argue that when the teacher is wrong, it is better to increase the weight of the distillation loss and increase the weight of the classification loss (assuming training class labels are known at distillation time).

The authors perform experiments on CIFAR-100 and on ImageNet, do a number of ablation experiments, and provide comparisons to many existing knowledge-distillation algorithms. The suggested approach does show a consistent improvement, albeit arguably small.